# LLM-BASED ONLINE TIME SERIES FORECASTING WITH FREQUENCY-DRIVEN PATTERN RECOGNITION

## ABSTRACT

Online Time Series Forecasting (OTSF) task has been consistently studied due to its practicality in multiple domains. Considering the sequential and evolving nature of time series, OTSF models must be robust to distribution shifts and possess long-term adaptability for practical scenarios. However, existing research falls short due to lack of explicit handling of time series patterns and limitations of memory buffer-based retrieval strategies. In this paper, we propose a novel LLM-based online time series forecaster, called LLM4OT, which excels not only in continuous distribution shifts, but also in extended online scenarios. Our main idea can be summarized in two points: (1) By representing time series as a combination of frequency bases, and encoding the knowledge of each basis into prompts that guide the data distribution, our model can effectively adapt to unobserved patterns. (2) By collaboratively employing pretrained LLM with time series backbone, we enhance the model's adaptation to data-scarce online scenarios. Additionally, we provide text-based descriptions that the LLM can easily understand, enriching the sparse data and maximizing the LLM's adapting ability without requiring training. Our extensive experiments on various real-world datasets demonstrate superiority and practicality of LLM4OT in various scenarios, including cross-dataset scenarios that maximize distribution shifts and scenarios with an extended online phase. Our code is available at `https://anonymous.4open.science/r/LLM4OTSF-38FE/`.

## 1 INTRODUCTION

Early research in deep learning-based time series forecasting(Nie et al., 2022; Zhang & Yan, 2023; Zhou et al., 2022b; 2021; Wu et al., 2021; 2022) primarily focused on batch learning methods utilizing static training and evaluation datasets. However, given the sequential and evolving nature of time series data, shifts in underlying patterns over time are inevitable. Consequently, traditional batch learning approaches often fail to adapt to such changes, while frequent model retraining to accommodate new patterns is both labor-intensive and impractical for real-world applications. To address these challenges, online learning paradigms, which enable models to incrementally update as new data arrive in dynamic environments, have been increasingly explored.

Among existing studies on online time series forecasting (OTSF) (Pham et al., 2022; Wen et al., 2023; Lau et al., 2025), the pioneering work FSNet (Pham et al., 2022) highlighted and tackled the following two key challenges in the OTSF task: (1) The model should be capable of rapidly adapting to continuously arriving data with only a few training steps. This is addressed by incorporating lightweight per-layer adapters that directly modify parameters and features, enabling swift adaptation at each layer. (2) The model should be capable of retaining and reusing previously learned patterns to respond effectively when similar patterns reoccur. This is addressed by introducing an associative memory that can store, update, and retrieve recurring patterns. Building on FSNet, subsequent studies (Wen et al., 2023; Lau et al., 2025) further improved adaptation capabilities. OneNet (Wen et al., 2023) noted that different modeling strategies have distinct advantages depending on the target dataset and time points, and thus employed reinforcement learning to dynamically learn ensemble weights that optimize model selection over time. Additionally, DSOF (Lau et al., 2025) identified issues of information leakage present in prior online learning setups and proposed a dual-stream (i.e., teacher-student) residual framework to address delayed model adaptation.

Although aforementioned studies have effectively tackled the OTSF task, several fundamental challenges remain unresolved. **(1) Inadaptability to unseen patterns.** In online forecasting scenarios, previously unseen patterns (i.e., distribution shifts in pattern) may emerge during test time (i.e., online phase). Thus, models must posses adaptability to such shifts for effective adjustments. Figure 1(a) shows experimental results obtained under a setting where the time series patterns in the training and online phases are intentionally made different. Specifically, we used two datasets from the same domain (i.e., ETTh1 and ETTh2) and conducted experiments under two scenarios: One where model

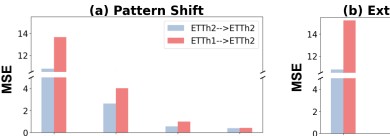 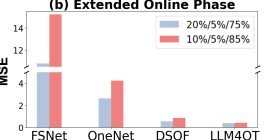 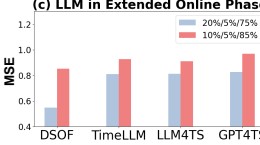

Figure 1: (a) Performance of prior methods and our proposed method (i.e., LLM4OT) in a cross-dataset scenario with distribution shifts in pattern. (b) Performance of prior methods and LLM4OT under an extended online phase. (c) Performance of DSOF and LLM-based time series forecasting methods under an extended online phase (ETTh2 dataset is used for (b) and (c)).

training and online updates are both performed on ETTh2 (in blue), and the other where models are initially trained on ETTh1 dataset, then updated online as ETTh2 data streams in[1] (in red). We observe that existing OTSF methods degrade substantially on patterns unseen during training, as they rely on associative memory to retrieve patterns similar to the new input. In other words, when an unprecedented pattern appears, retrieval from previously learned patterns becomes unreliable, leading to failure in adaptation under distribution shifts. **(2) Inadaptability to extended online phases.** Since new data are continuously streamed in online scenarios, models need enough capacity for long-term adaptation to effectively learn from evolving patterns. Figure 1 (b) shows that increasing the proportion of the online phase (i.e., test data) leads to significant performance deterioration in previous approaches[2]. This degradation stems from two primary issues: (i) The increasing diversity of recurring patterns makes it impractical to store them all explicitly in the associative memory, which can lead to catastrophic forgetting, and (ii) the emergence of previously unseen patterns exacerbates the model's limited adaptability.

To address the challenges of adaptability to unseen patterns and extended online phases, we propose LLM4OT, an Large Language Model (LLM)-based online time series forecasting framework built upon two key principles. First, to improve the model's ability to adapt to unseen patterns, we guide it with frequency-domain information through a prompting method (Kim et al., 2023). Since frequency representations are effective for capturing periodic structures that are often hard to detect in the time domain (Zhou et al., 2022b; Yi et al., 2023), we decompose the input series into frequency components using the Discrete Fourier Transform (DFT), where each frequency basis is assigned a learnable embedding, scaled by its amplitude, and fed to the model as prompts. This design allows the model to exploit frequency-domain cues and more effectively recognize time series patterns required for OTSF. This framework provides the following advantages for OTSF: (i) *Effectiveness*: Newly emerged unseen patterns can be effectively represented by combining the learned frequency bases, unlike existing methods that struggle to retrieve newly emerged unseen patterns from an associative memory. (ii) *Efficiency*: Whereas existing methods that continuously store observed patterns eventually exhaust memory, representing patterns as combinations of frequency bases avoids memory requirements and allows the model to scale as the number of patterns increases in the online phase without additional training.

Second, to maintain adaptability over extended online phases, we integrate a pre-trained LLM with a time series backbone. Recent studies (Jin et al., 2023; Chang et al., 2023; Zhou et al., 2023) have shown that LLMs can significantly enhance forecasting performance when aligned with time series tasks, particularly in few-shot or zero-shot settings. Such transferability of LLMs, achievable without training, is especially advantageous in OTSF, as it enables effective and efficient adaptation under limited data. In Figure 1 (c), we compare the state-of-the-art OTSF model that relies solely on a time series model without LLMs (i.e, DSOF) against those that align the time series backbone with LLMs (i.e., TimeLLM (Jin et al., 2023), LLM4TS (Chang et al., 2023), and GPT4TS (Zhou et al., 2023)) under an extended online phase. Surprisingly, we found that although these LLM-based models were not originally designed for online learning[3], they demonstrate performance comparable to DSOF as the online phase lengthens. These results demonstrate that, as the online phase progresses and becomes increasingly data-scarce, the transferability of LLMs proves to be particularly effective. To further enhance the adaptability of LLMs in the OTSF task, we generate text-based descriptions as additional information that summarize recent patterns from both time and frequency domain perspectives. Specifically, recent patch values are directly provided as the time domain information, while we adopt Discrete Wavelet Transform (DWT) as the frequency domain information. Note

---

[1]A wider range of scenarios is provided in Table 3

[2]To create a more challenging and extended online phase within the given dataset, we deviate from the typical 20%/5%/75% split used in prior studies and instead adopt a 10%/5%/85% train/validation/test split.

[3]We adhere to their original training protocols and fine-tune only the output projection in the online phase.

that we use DWT instead of DFT as Wavelet Transform enables localized time-frequency analysis, making it more suitable for capturing *recent* frequency variations essential for online adaptation. Incorporating these textual descriptions supports stable and rapid adaptation, even as the online phase extends, for two key reasons: (1) The additional semantic information complements the limited training data, enhancing model generalization, as reflected in the improved performance shown in Figure 1 (b). (2) Since the descriptions are provided without requiring additional model training, they offer an efficient mechanism for adaptation without incurring computational overhead.

In this study, we make the following contributions:

- We identify that existing OTSF methods struggle to effectively capture ongoing distribution shifts, particularly due to their vulnerability to patterns that were unseen during the training phase. To address this limitation, we introduce a prompting method that incorporates frequency-domain bases to explicitly represent pattern information and improve adaptation to unseen patterns.

- We present the first study to integrate LLMs into OTSF, utilizing their rich knowledge and transferability to enable effective adaptation in continuously evolving, data-scarce online scenarios.

- Through extensive experiments under various online learning scenarios, we demonstrate that LLM4OT consistently outperforms state-of-the-art OTSF methods.

## 2 RELATED WORK

We provide a concise overview of related work in this section. A complete discussion is in Appendix A.

**Online Time Series Forecasting.** Given the evolving nature of time series data, online forecasting has gained prominence for practical applications (Kuznetsov & Mohri, 2016; Gultekin & Paisley, 2018; Aydore et al., 2019). Recently, online deep learning models have been proposed to further capture complex patterns within time series data. FSNet (Pham et al., 2022) introduces calibration module to dynamically balance fast adaptation to recent changes with the retention of prior knowledge. OneNet (Wen et al., 2023) incorporates reinforcement learning to model cross-variable and cross-time concept drifts. Addressing the information leakage issue in previous research, DSOF (Lau et al., 2025) redefines the OTSF setting and proposes a dual-stream mechanism to update model parameters. Nevertheless, prior studies do not explicitly model the patterns in input signals, hindering their ability to adapt to unobserved distributions. Additionally, their explicit storage of pattern information restricts the model's adaptability to extended online phases.

**Prompt-based Continual Learning.** Rehearsal-free continual learning methods leverage the strong general representations of pre-trained models like ViT (Dosovitskiy et al., 2020). By fine-tuning only small, learnable *prompts* for each task, these methods achieve significant memory and computational efficiency, as the core model parameters remain unchanged. VPT (Jia et al., 2022) optimizes a single prompt, L2P (Wang et al., 2022c) uses a shared pool of prompts. S-Prompts (Wang et al., 2022a), train a unique prompt for each individual task to address catastrophic forgetting. However, the application of prompt learning to address distribution shifts in the time series domain remains unexplored. Given the continuous nature of time series data, the online learning scenario is more suitable than continual learning, which assumes distinct tasks. LLM4OT is the first to achieve an efficient and scalable prompting crucial for online learning scenarios by encoding knowledge from the underlying frequency bases.

**Frequency Analysis in Time Series Forecasting.** Due to the complex temporal variations in time series data, frequency analysis techniques like Discrete Fourier Transform (DFT) and Discrete Wavelet Transform (DWT) are used to capture recurring patterns. DFT analyzes global frequency components, while DWT provides localized frequency information at different scales. FEDformer (Zhou et al., 2022b) proposes two possible structures, Fourier Enhanced Structure and Wavelet Enhanced Structure, each leveraging DFT and DWT respectively. While prior studies model frequency domain dependencies effectively, their reliance on coefficient values alone, without learning the underlying knowledge of each basis, limits robustness to distribution shifts. In contrast, LLM4OT can express unobserved pattern by a combination of basis embeddings by representing time series patterns as a combination of learnable frequency bases. This enhances adaptability to distribution shifts, making it more suitable for online scenarios.

**Time Series Forecasting with LLMs.** Recent advancements in LLMs have prompted researchers to investigate their transferability to forecasting tasks in data-sparse time series domains. LLM4TS (Chang et al., 2023) introduces two-stage fine-tuning approach to leverage LLMs for time series forecasting. GPT4TS (Zhou et al., 2023) retrains the positional embeddings and normalization layers

of LLMs to preserve pre-trained knowledge while enhancing performance on downstream tasks. Additionally, TimeLLM (Jin et al., 2023) employs reprogramming method to align time series data with word embeddings. Inspired by the proven adaptability of these models, we leverage LLMs to enable rapid adjustments in online scenarios.

# 3 PRELIMINARIES

## 3.1 TIME SERIES FORECASTING

Let $\mathbf{X} = (x_1, \ldots, x_{N_{data}}) \in \mathbb{R}^{N_{data} \times n}$ be the entire time series with $N_{data}$ observations, where each observation $x_i \in \mathbb{R}^n$ contains $n$ dimensions. The dataset $\mathbf{X}$ is then partitioned into $N_{train}$, $N_{val}$, and, $N_{online}$ time stamps according to predefined ratios, maintaining the chronological order of the data. Given the look-back window of length $L$, denoted as $\mathbf{X}_{i-L+1:i} = (x_{i-L+1}, x_{i-L+2}, \ldots, x_i)$, the objective of time series forecasting is to predict the following $H$ steps (i.e., $\mathbf{X}_{i+1:i+H}$), where the model's prediction at $t = i$ for the next $H$ steps is denoted by $\hat{\mathbf{X}}_{i+1:i+H} = (\hat{x}_{i+1}, \hat{x}_{i+1}, \ldots, \hat{x}_{i+H}) = f(\mathbf{X}_{i-L+1:i})$. The objective is to minimize the mean squared error (MSE) between the ground truth and the predicted outputs, i.e., $\frac{1}{H}\Sigma_{h=1}^{H}||\hat{x}_{i+h} - x_{i+h}||_2^2$.

## 3.2 ONLINE TIME SERIES FORECASTING

The OTSF scenario consists of two phases: training phase and online phase. In the training phase, the entire $N_{train}$ time series are utilized to create $(L + H)$ sized time sequences. The objective of the training phase is to let the model to learn the base knowledge through static batch training strategy. The online phase follows the training phase, where $N_{online}$ time steps are streamed sequentially with a moving window of size 1. This mirrors real-world scenarios, and the model is updated in real-time.

**Objective and Evaluation Criterion.** Our ultimate goal is to accurately predict the ground truth by minimizing the cumulative MSE between the ground truth and predicted values over the entire prediction horizon of $H$ steps, using $\text{MSE}_{online}$ to evaluate performance as follows:

$$\text{MSE}_{online} = \frac{1}{N_{online} - L - H + 1} \sum_{i=N_{train}+N_{val}+L}^{N_{data}-H} ||f(\mathbf{X}_{i-L+1:i}) - \mathbf{X}_{i+1:i+H}||_2^2. \tag{1}$$

# 4 PROPOSED METHOD: LLM4OT

In this section, we introduce our proposed method LLM4OT. The key components of this framework are as follows: (1) explicitly learning pattern embeddings by decomposing time series patterns into frequency bases (Sec. 4.1), and (2) collaboratively utilizing a pre-trained LLM with the time series backbone with textual recent information that enhances the model's adaptability in data-scarce and rapid-tuning online scenario (Sec. 4.2). Overall framework of LLM4OT is shown in Figure 2.

## 4.1 CAPTURING TIME SERIES PATTERNS VIA FREQUENCY DOMAIN ANALYSIS

As the online scenario progresses, time series patterns continuously evolve and new patterns emerge. However, prior methods (Pham et al., 2022; Wen et al., 2023; Lau et al., 2025) that retrieve previously learned knowledge from an associate memory cannot address such newly emerging patterns. To overcome this limitation, our framework first decomposes time series into its underlying frequency bases rather than storing it directly, and then encodes this basis-level knowledge into prompts. This approach allows new patterns to be represented as compositions of the learned bases. Frequency-based analysis enables effective decomposition of time series data into basis sequences that capture distinct patterns, as frequency components are intrinsically associated with the underlying temporal patterns that are often challenging to analyze in the time domain. Specifically, we use the DFT to decompose the input time series $\mathbf{X}$ of length $L$. The DFT converts the sequence from the time domain to the frequency domain, while its inverse (IDFT) converts it back. Their expressions are as follows:

$$\mathcal{F}(k) = DFT(\mathbf{X}) = \sum_{n=0}^{L-1} \mathbf{X}[n]\exp\left(-i\frac{2\pi kn}{L}\right), \quad k = 0, 1, \ldots, L-1, \tag{2}$$

$$\mathbf{X}[n] = IDFT(\mathcal{F}) = \frac{1}{L} \sum_{k=0}^{L-1} \mathcal{F}(k)\exp\left(i\frac{2\pi kn}{L}\right), \quad n = 0, 1, \ldots, L-1, \tag{3}$$

where $\mathcal{F}$ refers to the frequency spectrum of the input, $\mathbf{X}[n]$ refers to the $n$-th index in the time series $\mathbf{X}$, and $i$ represents the imaginary unit. From the perspective of frequency basis, assuming that $L$ is even, both the DFT and IDFT can be represented using $\frac{L}{2} + 1$ orthogonal cosine-based frequency

basis because the DFT of a real-valued signal exhibits Hermitian symmetry. Thus, the IDFT can be rewritten as follows:

$$\mathbf{X}[n] = \frac{1}{L} \sum_{k=0}^{\frac{L}{2}} \left( \mathbf{R}_k \cdot \cos\left(\frac{2\pi kn}{L} - \phi\right) \right) = \frac{1}{L} \sum_{k=0}^{\frac{L}{2}} \left( \mathbf{R}_k \cdot \text{freq}_k \right), \quad n = 0, 1, \ldots, L-1, \tag{4}$$

where $\text{freq}_k$ and $\mathbf{R}_k \in \mathbb{R}$ denote the basis of the $k$-th frequency and its amplitude, respectively.

Since a time series can be expressed as a combination of frequency bases using the DFT, our next goal is to provide the model with the relevant bases and their amplitudes for effective pattern recognition. To achieve this, we utilize a prompting method, which involves adding small learnable parameters, known as prompts, to the input data in order to refine the model. More specifically, a prompt is assigned to each frequency basis, and this prompt learns the knowledge associated with the corresponding frequency basis, i.e., $\mathbf{P} = [\mathbf{p}_0, \ldots, \mathbf{p}_{\frac{L}{2}}] \in \mathbb{R}^{(\frac{L}{2}+1) \times d}$, where $\mathbf{P}$ is the prompt bank containing the prompts corresponding to each frequency basis, and $\mathbf{p}_k$ is the prompt corresponding to the $k$-th frequency basis. However, learning knowledge for all frequency basis is not effective in capturing the overall pattern of the given time series. That is, high frequencies represent rapidly oscillating periodicities compared to low frequencies, and therefore, they do not capture the overall pattern information. Hence, in time series analysis, high frequencies are often treated as noise, which is why low-pass filtering techniques (Zhou et al., 2022a; Xu et al., 2023) are widely used. Accordingly, to effectively capture the overall pattern while removing noise, we introduce a hyperparameter $\gamma \in [0, 1]$ to eliminate the high-frequency basis, i.e., $\mathbf{P}_{low} = [\mathbf{p}_0, \ldots, \mathbf{p}_{\lceil \frac{L}{2} \cdot \gamma \rceil}] \in \mathbb{R}^{\lceil (\frac{L}{2} \cdot \gamma + 1) \rceil \times d}$. We then generate a pattern embedding, which serves as a prompt to inform the model of the distribution (i.e., pattern), by concatenating each prompt weighted by the amplitude of its corresponding frequency basis as follows:

$$\mathcal{P}_{\mathbf{X}} = \text{Concat}\left( \mathbf{R}_0 \cdot \mathbf{p}_0, \mathbf{R}_1 \cdot \mathbf{p}_1, \ldots, \mathbf{R}_{\lceil \frac{L}{2} \cdot \gamma \rceil} \cdot \mathbf{P}_{\lceil \frac{L}{2} \cdot \gamma \rceil} \right) \in \mathbb{R}^{\lceil (\frac{L}{2} \cdot \gamma + 1) \rceil \times d}, \tag{5}$$

where $\mathcal{P}_{\mathbf{X}}$ refers the pattern embedding of time series $\mathbf{X}$ to be provided to the model. Through frequency basis-based pattern analysis and embedding calculation, the model can explicitly capture pattern information in scenarios where distribution shifts continuously occur. Even when a previously unseen pattern emerges, the model can represent and adapt to the pattern by combining the knowledge learned from each frequency basis[4]. Moreover, since newly emerging patterns can be expressed as combinations of a finite set of basis, the model maintains its memory efficiency without degradation.

## 4.2 ENHANCING MODEL ADAPTABILITY USING A PRE-TRAINED LLM

In this section, we propose a strategy that leverages a pre-trained LLM with a time-series backbone network for effective, rapid adaptation in data-scarce online scenarios. Beyond simply aligning the LLM with the time series backbone, we further enhance LLM's ability to adapt to changing patterns by providing a description of the recent pattern in a text format that the LLM can easily understand. Given the input time series $\mathbf{X}$, the text description (i.e., $text_{\mathbf{X}}$) contains task details, a description of the dataset, and details about the recent pattern from both the time- and frequency-domain perspectives. From the time domain perspective, the actual values of the most recent patch are provided to convey information about the recently occurred time variant. From the frequency domain perspective, the objective is to provide the frequency information of the most recent time point. This differs from the goal in Section 4.1, where frequency component analysis is used to capture the overall pattern. Thus, it is important to note that, in addition to frequency information, *time-related information* must also be captured. A naive approach for capturing recent frequency information would be to adopt the Short-Time Fourier Transform (STFT), which, unlike the DFT that provides a global frequency representation over the entire time series, applies a fixed window to enable localized frequency analysis over time. However, due to its fixed window size, STFT suffers from a trade-off between time and frequency resolution, i.e., uncertainty principle (Cohen, 1995), making it unsuitable for scenarios with continuously changing patterns. This limitation prevents it from effectively capturing evolving, non-stationary signals, as it lacks the temporal resolution needed for transient events and high-frequency variations. Hence, we utilize the Discrete Wavelet Transform (DWT), which offers greater flexibility compared with the STFT in capturing time-varying patterns. Unlike STFT's fixed window with limited temporal resolution, DWT adapts window size by frequency, providing fine resolution at high frequencies and broader at low frequencies, thus better capturing recent frequency information in non-stationary signals. Given the input time series $\mathbf{X}$, the

---

[4]In Appendix F.5, we observe that the frequency basis-driven prompt bank, trained only during the training phase, can effectively adapt to unseen patterns that emerge in the online phase.

equation of DWT using a scaling function $\phi$ and a wavelet function $\psi$ is as follows:

$$\mathbf{A}_j[k] = \sum_n \mathbf{X}[n]\phi_{j,k}[n], \quad \mathbf{D}_j[k] = \sum_n \mathbf{X}[n]\psi_{j,k}[n], \tag{6}$$

where $\mathbf{A}_j[k]$ and $\mathbf{D}_j[k]$ refer to the approximation and detail coefficient at level $j$, respectively, and $\phi_{j,k}[n]$ and $\psi_{j,k}[n]$ are the scaling and wavelet functions at level $j$, respectively. After the decomposition, the time series is passed through a filter bank that separates the low-pass and high-pass components, and downsampling is performed as follows:

$$\mathbf{A}_{j+1}[k] = \sum_n h[n - 2k]\mathbf{A}_j[n], \quad \mathbf{D}_{j+1}[k] = \sum_n g[n - 2k]\mathbf{A}_j[n], \tag{7}$$

where $h[n]$ and $g[n]$ refer the low- and high-pass filters, respectively. Through this process, we utilize $\mathbf{A}_j[-1]$ to provide the model with the most recent frequency information where $j$ is a hyperparameter.

An example of the text description is shown in Figure 2. The text description is first processed through the pre-trained LLM's tokenizer. The resulting token IDs are then passed through the LLM's frozen input embedding layer to retrieve their corresponding token embeddings. This sequence of text token embeddings is the resulting embedding (denoted as $\mathcal{T}_\mathbf{X}$) that is used as input for the final prediction. Providing text descriptions enriches data in data-scarce online scenarios, offering recent information that aids effective adaptation, all without requiring additional training.

## 4.3 OVERALL FRAMEWORK

Figure 2 shows the overall framework, and the pseudo code can be found in Appendix G.

**Align Module.** In OTSF scenarios, effectively aligning continuous time series data with discrete token-based LLMs is crucial yet challenging. Since pre-trained LLMs lack inherent knowledge of time series patterns, prior studies (Jin et al., 2023; Chang et al., 2023; Zhou et al., 2023) focus on aligning two modalities (i.e., time series and language) to leverage the knowledge within LLMs, enabling accurate, data-efficient, and task-agnostic forecasting. As the goal of this study is to enhance the LLM's ability in the online scenario, not to focus on the align module itself, we utilize a pre-developed align module (Jin et al., 2023; Chang et al., 2023). The align module aligns the representation of the time series computed by the time series backbone (i.e., $emb_\mathbf{X} = b(\mathbf{X})$, where $b(\cdot)$ is the time series backbone) with the natural language modality, and outputs the

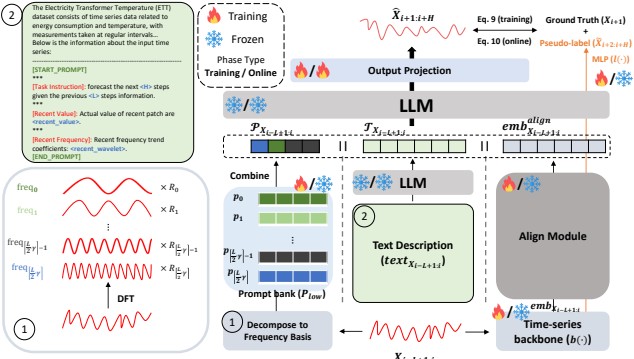

Figure 2: Overall model framework. Given the input time series $\mathbf{X}_{i-L+1:i}$, the aligned embedding (i.e., $emb_{\mathbf{X}_{i-L+1:i}}$), embedded text description (i.e., $\mathcal{T}_{\mathbf{X}_{i-L+1:i}}$), and pattern embedding (i.e., $\mathcal{P}_{\mathbf{X}_{i-L+1:i}}$) are provided as input to the LLM. The representation computed by the LLM is passed through the output projection layer to produce the final prediction, i.e., $\hat{\mathbf{X}}_{i+1:i+H}$.

resulting aligned embeddings (i.e., $emb_\mathbf{X}^{align}$) [5]. As shown in Appendix E.2, the application of various align modules consistently resulted in strong performance, indicating that LLM4OT allows for the agnostic application of the align module.

**Utilizing a pre-trained LLM.** The input to the LLM is formed by concatenating the pattern embedding (i.e., $\mathcal{P}$), embedded text description (i.e., $\mathcal{T}$), and aligned time series embedding (i.e., $emb^{align}$). The concatenated embedding sequence is input to the LLM's transformer layers, all of which remain frozen. The last hidden state representation of the LLM serves as the time series representation, which is flattened and linearly projected to generate the final forecast.

**Training Phase.** The training phase serves to learn the overall base knowledge before entering the online phase. Therefore, during the training phase, except for the pre-trained LLM, we train the time series backbone network, align module, prompt bank (i.e., learnable prompts for each frequency basis), and output projection layer by minimizing the following objective:

$$\mathcal{L}_{training} = \frac{1}{N_{train} - L + 1} \sum_{i=L}^{N_{train}} ||f(\mathbf{X}_{i-L+1:i}) - \mathbf{X}_{i+1:i+H}||_2^2. \tag{8}$$

where $f(\cdot)$ is the overall framework of LLM4OT.

---

[5]For detailed information on the operation of each align module, please refer to Appendix E.

**Online Phase.** In the online phase, only the output projection layer is tuned to match the streaming data distribution, while all other parameters, including the pre-trained LLM, time series backbone network, align module, and prompt bank, remain frozen[6]. According to (Lau et al., 2025), when the prediction horizon $H$ is greater than 1, calculating the loss using the ground truth for all $H$ time steps ($\mathbf{X}_{i+1:i+H}$) at each moving window step for model updates leads to information leakage. To avoid this, the moving window should be extended to $H$ steps rather than updating at each step, which however introduces an update delay and hinders effective adaptation. Hence, we employ a pseudo-labeling technique while keeping updates at each step. When calculating the loss for the model's output at $t = i$ to update the model, only the ground truth for the immediate next time point, i.e., $\mathbf{X}_{i+1}$ is used, and pseudo-labels are generated and utilized for the remaining time steps (i.e., $\tilde{\mathbf{X}}_{i+2:i+H}$). We propagate the representation of the time series backbone network (i.e., $b(\cdot)$) through a linear layer (i.e., $l(\cdot)$) to project it into the output space and obtain the pseudo-label. This linear layer is utilized while being frozen during the online phase, and is trained along with Equation 8 during the training phase using MSE loss. Therefore, Equation 8 is modified as follows:

$$\mathcal{L}_{training}^{+} = \frac{1}{N_{train} - L + 1} \sum_{i=L}^{N_{train}} \left( ||f(\mathbf{X}_{i-L+1:i}) - \mathbf{X}_{i+1:i+H}||_2^2 + ||l(b(\mathbf{X}_{i-L+1:i})) - \mathbf{X}_{i+1:i+H}||_2^2 \right), \quad (9)$$

where $l(\cdot)$ denotes the pseudo-label projection layer which projects the time series representation into the output space. During the online phase, we use the frozen $b(\cdot)$ and $l(\cdot)$, which are trained in the training phase, to generate pseudo-labels: $\tilde{\mathbf{X}}_{i+2:i+H} = l(b(\mathbf{X}_{i-L+1:i}))[1:]$. These pseudo-labels are then employed to tune the model. To mitigate the impact of prediction errors and pseudo-labels as the forecast horizon extends from the current observation, we apply a geometric decay factor $\delta \in [0, 1]$ to the online loss as follows:

$$\mathcal{L}_{online} = \frac{1}{N_{online} - L - H + 1} \sum_{i=N_{train}+N_{val}+L}^{N_{data}-H} \left( \frac{1}{H} \sum_{h=1}^{H} \delta^{h-1} ||\hat{\mathbf{X}}_{i+h} - \overline{\mathbf{X}}_{i+h}||_2^2 \right), \quad (10)$$

where $\overline{\mathbf{X}}_{i+1:i+H} = \text{Concat}(\mathbf{X}_{i+1}, \tilde{\mathbf{X}}_{i+2:i+H})$ is a concatenated sequence of the ground truth and pseudo-labels. Finally, the model's online performance is evaluated using Equation 1.

## 5 EXPERIMENTS

**Datasets.** Following prior studies (Pham et al., 2022; Wen et al., 2023; Lau et al., 2025), we utilize datasets from various domains (i.e., ETT, Weather, ECL, and Traffic), splitting the time series data into training, validation, and testing sets with a 20%, 5%, and 75% split, respectively. Please refer to Appendix B for dataset details.

**Baselines.** We utilize various deep learning-based time series forecasting models as baselines. **DLinear** (Zeng et al., 2023), **PatchTST** (Nie et al., 2022), **iTransformer** (Liu et al., 2023b), and **TimeMixer** (Wang et al., 2024b) developed in a static time series forecasting scenario, LLM-based time series forecasting models (i.e., **LLM4TS** (Chang et al., 2023), **GPT4TS** (Zhou et al., 2023), and **Time-LLM** (Jin et al., 2023)), and prior research on OTSF scenarios (i.e., **FSNet** (Pham et al., 2022), **OneNet** (Wen et al., 2023), and **DSOF** (Lau et al., 2025)) are used as baselines. Please refer to Appendix C for baseline details.

**Implementation Details.** Consistent with previous studies (Pham et al., 2022; Wen et al., 2023; Lau et al., 2025), we set prediction length $H$ to 1, 24, and 48, with a lookback length $L$ of 96. We utilize PatchTST (Nie et al., 2022) as time series backbone network $b(\cdot)$ and Llama-7B (Touvron et al., 2023) as the default LLM unless stated otherwise. The evaluation metrics include mean square error (MSE) and mean absolute error (MAE). Please refer to Appendix D for implementation details.

### 5.1 OVERALL PERFORMANCE

The experimental results on seven datasets are summarized in Table 1. The reported results represent averages over three runs, with standard deviations detailed in Table 6 of Appendix F. We make the following key observations: **(1)** In settings without information leakage[7], FSNet and OneNet, which are designed for OTSF, underperform static models such as DLinear, iTransformer, and TimeMixer. This implies that FSNet and OneNet mainly exploit the leaked data on which they are trained—leading to rapid convergence—rather than truly acquiring the underlying structure of newly arriving patterns or learning how to adapt to them. **(2)** LLM-based models, i.e., LLM4TS, GPT4TS, and Time-LLM,

---

[6]We emphasize that LLM4OT is an efficient framework despite utilizing an LLM, as the number of parameters updated during the online phase is extremely small. Please see Appendix F.2 for a detailed analysis.

[7]Please refer to Appendix F.9 for experimental results under the information leakage setting used by FSNet and OneNet, two of our major baselines.

Table 1: Comparison of MSE and MAE results in OTSF for predicting 1, 24, and 48 prediction horizon (i.e., $H$) with a lookback length $L = 96$ (Best: bold red, the second-best: underlined in blue).

| | H | DLinear MSE | MAE | PatchTST MSE | MAE | iTransformer MSE | MAE | TimeMixer MSE | MAE | LLM4TS MSE | MAE | GPT4TS MSE | MAE | Time-LLM MSE | MAE | FSNet MSE | MAE | OneNet MSE | MAE | DSOF MSE | MAE | LLM4OT MSE | MAE |
|---|---|---|---|---|---|---|---|---|---|---|---|---|---|---|---|---|---|---|---|---|---|---|---|
| ETTh1 | 1 | 0.502 | 0.609 | 0.779 | 0.828 | 0.993 | 0.976 | 0.557 | 0.706 | 1.436 | 1.058 | 1.539 | 1.141 | 1.382 | 1.076 | 13.26 | 3.441 | 4.023 | 1.956 | 0.802 | 0.857 | 0.425 | 0.601 |
| | 24 | 2.333 | 1.427 | 3.797 | 1.849 | 3.028 | 1.690 | 2.719 | 1.528 | 3.155 | 1.623 | 3.028 | 1.640 | 2.947 | 1.417 | 19.33 | 4.097 | 9.001 | 2.801 | 2.311 | 1.372 | 1.217 | 0.993 |
| | 48 | 2.802 | 1.553 | 5.135 | 2.066 | 3.998 | 1.929 | 3.274 | 1.699 | 6.256 | 2.312 | 6.738 | 2.397 | 5.997 | 2.049 | 25.82 | 4.881 | 12.21 | 3.334 | 4.233 | 1.973 | 1.482 | 1.073 |
| ETTh2 | 1 | 0.468 | 0.643 | 0.895 | 0.924 | 0.874 | 0.888 | 0.439 | 0.563 | 0.811 | 0.875 | 0.827 | 0.829 | 0.810 | 0.836 | 10.78 | 3.083 | 2.634 | 1.531 | 0.549 | 0.537 | 0.398 | 0.423 |
| | 24 | 2.187 | 1.338 | 4.885 | 2.002 | 2.688 | 1.451 | 1.938 | 1.296 | 3.023 | 1.538 | 2.887 | 1.509 | 2.381 | 1.503 | 17.99 | 4.041 | 7.019 | 2.338 | 2.179 | 1.410 | 0.932 | 0.607 |
| | 48 | 2.349 | 1.412 | 6.368 | 2.349 | 3.768 | 1.541 | 2.216 | 1.398 | 5.411 | 2.026 | 5.510 | 2.237 | 5.231 | 2.087 | 22.79 | 4.673 | 9.930 | 3.019 | 4.003 | 1.991 | 1.245 | 0.699 |
| ETTm1 | 1 | 0.132 | 0.312 | 0.135 | 0.352 | 0.183 | 0.397 | 0.137 | 0.341 | 0.371 | 0.489 | 0.387 | 0.522 | 0.309 | 0.506 | 0.190 | 0.231 | 0.154 | 0.271 | 0.096 | 0.142 | 0.083 | 0.128 |
| | 24 | 0.618 | 0.771 | 1.102 | 1.029 | 1.101 | 1.009 | 0.602 | 0.726 | 0.702 | 0.707 | 0.711 | 0.741 | 0.671 | 0.795 | 1.520 | 1.202 | 1.103 | 1.003 | 0.412 | 0.452 | 0.372 | 0.403 |
| | 48 | 0.829 | 0.894 | 2.083 | 1.243 | 1.298 | 1.039 | 0.811 | 0.882 | 0.935 | 0.846 | 0.933 | 0.886 | 0.897 | 0.836 | 2.283 | 1.390 | 1.492 | 1.21 | 0.559 | 0.540 | 0.451 | 0.515 |
| ETTm2 | 1 | 0.111 | 0.303 | 0.113 | 0.296 | 0.137 | 0.350 | 0.114 | 0.326 | 0.302 | 0.525 | 0.321 | 0.536 | 0.187 | 0.333 | 0.127 | 0.326 | 0.111 | 0.297 | 0.066 | 0.219 | 0.052 | 0.208 |
| | 24 | 0.589 | 0.667 | 1.035 | 0.917 | 0.799 | 0.853 | 0.525 | 0.688 | 0.631 | 0.714 | 0.587 | 0.706 | 0.499 | 0.624 | 1.180 | 0.976 | 0.603 | 0.716 | 0.348 | 0.519 | 0.287 | 0.517 |
| | 48 | 0.809 | 0.794 | 1.802 | 1.202 | 1.003 | 0.950 | 0.784 | 0.854 | 0.775 | 0.833 | 0.713 | 0.749 | 0.589 | 0.707 | 1.847 | 1.194 | 0.843 | 0.881 | 0.348 | 0.628 | 0.348 | 0.539 |
| WTH | 1 | 0.359 | 0.499 | 0.521 | 0.691 | 0.491 | 0.607 | 0.338 | 0.543 | 0.489 | 0.659 | 0.493 | 0.602 | 0.488 | 0.649 | 0.731 | 0.754 | 0.501 | 0.536 | 0.298 | 0.398 | 0.052 | 0.114 |
| | 24 | 1.219 | 1.008 | 1.504 | 1.208 | 1.482 | 1.017 | 1.208 | 1.002 | 1.311 | 1.041 | 1.211 | 1.034 | 0.999 | 0.895 | 1.756 | 1.218 | 1.377 | 1.023 | 0.677 | 0.649 | 0.110 | 0.165 |
| | 48 | 1.741 | 1.247 | 2.001 | 1.257 | 1.699 | 1.146 | 1.733 | 1.216 | 1.610 | 1.169 | 1.651 | 1.185 | 1.483 | 1.078 | 2.620 | 1.609 | 2.703 | 1.591 | 0.919 | 0.813 | 0.215 | 0.297 |
| ECL | 1 | 2.911 | 1.617 | 4.279 | 1.968 | 1.897 | 1.232 | 2.708 | 1.549 | 6.354 | 2.371 | 6.384 | 2.327 | 6.045 | 2.355 | 311 | 16.18 | 29.88 | 4.99 | 4.639 | 2.141 | 0.160 | 0.220 |
| | 24 | 13.21 | 3.456 | 15.61 | 3.751 | 4.012 | 1.803 | 7.209 | 2.583 | 6.711 | 2.539 | 6.813 | 2.211 | 6.317 | 2.413 | 428 | 19.68 | 83.27 | 9.11 | 4.551 | 2.072 | 0.263 | 0.264 |
| | 48 | 25.98 | 4.906 | 15.88 | 3.849 | 4.877 | 2.008 | 9.243 | 2.840 | 7.342 | 2.596 | 7.523 | 2.474 | 7.321 | 2.557 | 469 | 19.65 | 144.89 | 11.43 | 5.819 | 2.355 | 0.344 | 0.317 |
| Traffic | 1 | 0.298 | 0.489 | 0.279 | 0.498 | 0.239 | 0.458 | 0.280 | 0.509 | 0.602 | 0.676 | 0.653 | 0.698 | 0.511 | 0.615 | 0.612 | 0.712 | 0.259 | 0.381 | 0.273 | 0.371 | 0.160 | 0.259 |
| | 24 | 0.656 | 0.704 | 0.591 | 0.668 | 0.458 | 0.646 | 0.661 | 0.796 | 0.697 | 0.735 | 0.711 | 0.803 | 0.636 | 0.747 | 0.759 | 0.811 | 0.581 | 0.589 | 0.369 | 0.386 | 0.307 | 0.281 |
| | 48 | 0.791 | 0.868 | 0.633 | 0.756 | 0.517 | 0.690 | 0.759 | 0.856 | 0.724 | 0.801 | 0.783 | 0.824 | 0.684 | 0.774 | 0.814 | 0.831 | 0.701 | 0.633 | 0.381 | 0.389 | 0.357 | 0.391 |

Table 2: Ablation studies of each component of LLM4OT(MSE).

| | $\mathcal{P}$ | $\mathcal{T}$ | ETTh2 1 | 24 | 48 | ETTm1 1 | 24 | 48 | WTH 1 | 24 | 48 | ECL 1 | 24 | 48 | Traffic 1 | 24 | 48 |
|---|---|---|---|---|---|---|---|---|---|---|---|---|---|---|---|---|---|---|
| (1) | ✗ | ✗ | 0.927 | 3.121 | 4.790 | 0.211 | 0.561 | 0.666 | 0.238 | 0.595 | 0.767 | 3.467 | 3.479 | 3.688 | 0.524 | 0.767 | 0.804 |
| (2) | ✓ | ✗ | 0.499 | 1.723 | 2.031 | 0.096 | 0.388 | 0.501 | 0.103 | 0.227 | 0.295 | 0.991 | 1.321 | 1.389 | 0.276 | 0.392 | 0.399 |
| (3) | ✗ | ✓ | 0.718 | 2.321 | 2.889 | 0.122 | 0.481 | 0.544 | 0.179 | 0.431 | 0.601 | 2.131 | 2.773 | 2.908 | 0.411 | 0.603 | 0.640 |
| (4)-1 | ✓ | ✓ | 0.398 | **0.932** | **1.245** | **0.083** | 0.372 | 0.451 | 0.052 | **0.110** | 0.215 | **0.160** | 0.263 | 0.344 | 0.211 | **0.307** | 0.357 |
| (4)-2 | ✓ | ✓(STFT) | **0.381** | 1.241 | 1.440 | 0.088 | 0.375 | 0.472 | **0.050** | 0.136 | 0.249 | 0.218 | 0.337 | 0.472 | **0.208** | 0.336 | 0.379 |

substantially underperform LLM4OT. This gap arises because their alignment modules cannot handle continuous distribution shifts, leading to a breakdown in modality alignment. Conversely, LLM4OT successfully adapts to these shifts using pattern embeddings and text descriptions, thereby preserving the crucial alignment between the LLM and the time series backbone. **(3)** While the state-of-the-art OTSF method, DSOF, demonstrates strong performance compared with other baselines, it significantly underperforms LLM4OT. This demonstrates that while the dual-stream framework of DSOF prevents update delays, allowing the model to adapt quickly without information leakage, its emphasis on rapid convergence to incoming data prevents it from understanding of underlying patterns, leaving the model poorly equipped to adapt when data are scarce. **(4)** LLM4OT demonstrates robust performance across multiple datasets by leveraging the adaptability of a pre-trained LLM in conjunction with a time series backbone.

## 5.2 ABLATION STUDY

To assess the impact of pattern embedding (i.e., $\mathcal{P}$) and text description (i.e., $\mathcal{T}$) in LLM4OT, Table 2 presents ablation studies across five cases, including the vanilla LLM4OT (Row (4)-1), with key observations as follows: **(1)** Introducing the pattern embedding is helpful (Row (1) vs. (2)). Given the sequential nature of time-series data, in which continuous distribution shifts are inevitable, providing the model with pattern embeddings that capture these dynamics is highly effective. Specifically, the frequency domain is utilized to capture the overall time series patterns, learning the knowledge of each frequency basis through decomposition. This enables effective adaptation to unseen patterns by combining the knowledge of basis components, resulting in nearly a 50% improvement in MSE. **(2)** Providing recent pattern information to the LLM in the form of text descriptions is effective (Row (1) vs. (3)). The text description provides additional information in data-scarce online scenarios without requiring any training, maximizing the few-shot transferability of LLMs and enabling effective adaptation. **(3)** Leveraging both pattern embedding and text description together can yield synergistic effects (Row (2&3) vs. (4)-1). Through pattern embeddings, the overall pattern of the given time series is captured, while text description provides information on recent patterns in the time and frequency domains, enabling the model to effectively adapt to recent patterns without being hindered by distribution shifts. **(4)** Utilizing DWT instead of STFT is more effective for providing recent pattern information from a frequency perspective (Row (4)-1 vs. (4)-2). STFT struggles with temporal resolution due to its use of a fixed window size, whereas DWT adapts the window size based on the frequency of the time series, making it more effective in capturing non-stationary signals. This results in comparable performance in relatively easy tasks with a prediction horizon of 1. However, for more challenging tasks with longer horizons, where capturing the underlying recent patterns is crucial, the use of DWT proves to be more effective. Additional ablation studies in more practical yet challenging cross-dataset and extended online scenarios is conducted in Appendix F.6.

Table 3: Comparison of MSE and MAE results in a cross-dataset scenario, where different datasets from the same ETT domain are used in the training and online phases to induce distribution shifts, with the prediction horizon (i.e., $H$) set to 1. $Training$ refers to the dataset used in the training phase, while $Online$ refers to the dataset used in the online phase.

| Training | Online | FSNet | | OneNet | | DSOF | | LLM4OT | | Online | FSNet | | OneNet | | DSOF | | LLM4OT | |
|---|---|---|---|---|---|---|---|---|---|---|---|---|---|---|---|---|---|---|
| | | MSE | MAE | MSE | MAE | MSE | MAE | MSE | MAE | | MSE | MAE | MSE | MAE | MSE | MAE | MSE | MAE |
| ETTh1 | →ETTh2 | 13.664 | 3.124 | 4.025 | 1.862 | 0.993 | 0.806 | **0.427** | **0.453** | →ETTm2 | 1.815 | 1.132 | 1.732 | 1.016 | 0.723 | 0.702 | **0.178** | **0.319** |
| ETTh2 | →ETTh1 | 12.667 | 3.011 | 3.833 | 1.808 | 0.901 | 0.849 | **0.399** | **0.531** | →ETTm2 | 1.791 | 1.029 | 1.489 | 1.022 | 0.818 | 0.744 | **0.211** | **0.359** |
| ETTm1 | →ETTh2 | 15.371 | 3.571 | 3.989 | 1.610 | 1.003 | 0.901 | **0.521** | **0.621** | →ETTm2 | 1.335 | 1.003 | 0.989 | 0.861 | 0.542 | 0.536 | **0.128** | **0.257** |
| ETTm2 | →ETTh2 | 14.989 | 3.199 | 4.138 | 1.734 | 1.211 | 0.945 | **0.513** | **0.616** | →ETTm1 | 1.299 | 1.039 | 1.315 | 0.946 | 0.517 | 0.619 | **0.132** | **0.263** |

Table 4: Comparison of MSE and MAE results across various datasets under scenarios with an extended online phase, where the train/valid/test split is set to 10%/5%/85%.

| | | ETTh2 | | | ETTm1 | | | WTH | | | ECL | | | Traffic | | |
|---|---|---|---|---|---|---|---|---|---|---|---|---|---|---|---|---|
| | | 1 | 24 | 48 | 1 | 24 | 48 | 1 | 24 | 48 | 1 | 24 | 48 | 1 | 24 | 48 |
| FSNet | MSE | 15.231 | 24.591 | 28.935 | 0.399 | 2.452 | 5.113 | 1.221 | 2.341 | 4.512 | 299.34 | 435.12 | 458.98 | 1.212 | 1.751 | 1.999 |
| | MAE | 3.802 | 4.758 | 5.179 | 0.516 | 1.365 | 2.061 | 1.004 | 1.330 | 2.041 | 15.301 | 17.859 | 20.423 | 1.001 | 1.232 | 1.353 |
| OneNet | MSE | 4.240 | 9.342 | 13.582 | 0.278 | 1.428 | 1.677 | 0.813 | 1.535 | 2.991 | 30.091 | 82.29 | 142.273 | 0.531 | 0.778 | 0.933 |
| | MAE | 2.009 | 3.006 | 3.485 | 0.427 | 1.074 | 1.204 | 0.801 | 1.098 | 1.599 | 5.285 | 8.041 | 10.797 | 0.588 | 0.742 | 0.905 |
| DSOF | MSE | 0.851 | 2.889 | 5.028 | 0.173 | 0.561 | 0.779 | 0.313 | 0.711 | 1.137 | 4.898 | 5.173 | 6.092 | 0.499 | 0.711 | 0.793 |
| | MAE | 0.872 | 1.499 | 2.042 | 0.397 | 0.668 | 0.846 | 0.539 | 0.803 | 1.006 | 2.134 | 2.204 | 2.368 | 0.656 | 0.759 | 0.801 |
| LLM4OT | MSE | **0.412** | **0.978** | **1.305** | **0.087** | **0.411** | **0.487** | **0.051** | **0.103** | **0.223** | **0.179** | **0.291** | **0.362** | **0.485** | **0.356** | **0.379** |
| | MAE | **0.434** | **0.613** | **0.730** | **0.194** | **0.601** | **0.637** | **0.206** | **0.291** | **0.452** | **0.403** | **0.509** | **0.581** | **0.434** | **0.481** | **0.578** |

## 5.3 FURTHER ANALYSIS

**Robustness to distribution shifts.** Table 3 shows the cross-dataset experiments using the ETT datasets. The ETTh series datasets (i.e., ETTh1 and ETTh2) are collected at an hourly interval, with ETTh1 representing a shorter time window and ETTh2 covering a longer period. The ETTm series datasets (i.e., ETTm1 and ETTm2), collected at 15-minute intervals, differ in feature count and time span, with ETTm2 providing finer temporal resolution compared to ETTm1. To induce distribution shifts, we deliberately use different datasets for the training and online phases. The model learns the base knowledge from the training data and adapts to the streaming online data, with the prediction horizon set to 1. We observe that LLM4OT outperforms all baselines across the 8 scenarios. FSNet, OneNet, and DSOF rely on associative memory to store recurring patterns and adapt by retrieving similar ones. This approach fails when unseen patterns arise, as no meaningful associations can be found, leading to substantial performance degradation (compare with the results in Table 1). In contrast, LLM4OT represents patterns through frequency bases and learns knowledge for each basis, rather than directly storing the time series patterns. As a result, even unseen patterns can be reliably expressed, making the model robust to distribution shifts and maintaining consistent performance. As a result, its performance remains largely consistent with the results in Table 1.

**Robustness to the extension of the online phase.** In Table 4, we analyze the model's robustness when the online phase is extended. Specifically, we evaluate performance by adjusting the OTSF train/valid/text split from 20%/5%/75% to 10%/5%/85%, extending the online phase [8]. Prior methods (i.e., FSNet, OneNet, and DSOF) experience significant performance degradation compared to the results in Table 1 for the following two reasons: (1) They are incapable of storing all the recurring patterns in an associative memory as the online phase is extended, and (2) the continuous occurence of unseen patterns prevents the model from maintaining adaptability. In contrast, LLM4OT, thanks to the rich knowledge and transferability of the pre-trained LLM, effectively maintains adaptability even in data-scarce online scenarios, demonstrating performance comparable to the results in Table 1. We argue that adaptability to an extended online phase is enhanced by two key designs: (i) a frequency basis-based embedding that generates efficient and reliable pattern representations from a finite set of bases, and (ii) a text description of recent patterns provided to the LLM, which compensates for data scarcity and enables efficient adaptation without additional training.

Adaptation efficiency during the online phase is further discussed in Appendix F.2. Additionally, Appendix F.3 and Appendix E.2 report performance across different backbone LLMs and alignment modules, respectively. Detailed sensitivity analysis and the impact of various component tuning strategies are presented in Appendix F.7 and Appendix F.8.

## 6 CONCLUSION

In this paper, we present the first LLM-based OTSF framework, called LLM4OT, which excels in both continuous distribution shift and extended online scenarios. We devise a strategy for generating

---

[8]To extend the online phase within the given dataset, we unavoidably adopt a 10%/5%/85% split setting. However, to isolate the effect of online phase length while keeping the training ratio fixed, we introduce an alternative setup in Appendix F.4 with a fixed 20% training split and a reduced 30% online phase.

pattern embeddings based on frequency bases to explicitly capture the overall pattern of the input sequence in online scenarios with continuous distribution shifts. This approach leverages the rich knowledge and superior transferability of LLMs to enable efficient rapid adaptation in data-scarce online scenarios. By providing a text description containing recent pattern information without requiring additional training, we enrich the data and maintain adaptation capabilities even as the online scenario extends. LLM4OT demonstrates promising performance across various real-world datasets, and exhibits robust performance against distribution shifts and the extension of the online phase, which highlights the applicability of LLM4OT in real-world scenarios.

## ETHICS STATEMENT

This work adheres to the ICLR Code of Ethics. All experiments were conducted on publicly available and widely used benchmark datasets (i.e., ETT, Weather, ECL, and Traffic), which do not contain personally identifiable or sensitive information, thus mitigating privacy concerns. Our work is designed to improve adaptability and efficiency in real-world forecasting tasks without introducing harmful applications or misuse. We are committed to scientific integrity and have made our anonymized source code available to ensure the transparency and reproducibility of our results. We have considered the potential implications of our work and conclude that it does not raise significant ethical issues.

## REPRODUCIBILITY STATEMENT

To ensure the reproducibility of our work, we provide all necessary details in Section 5 and Appendix D. Additionally, anonymized source code is available at `https://anonymous.4open.science/r/LLM4OTSF-38FE/`.

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

# Supplementary Material for
# LLM4OT: LLM-based Online Time Series Forecasting with Frequency-driven Pattern Recognition

# A  COMPLETE RELATED WORK

**Online Time Series Forecasting.**   Given the evolving nature of time series data, online forecasting has gained prominence for practical applications (Kuznetsov & Mohri, 2016; Gultekin & Paisley, 2018; Aydore et al., 2019). Unlike traditional methods, which separate the training and evaluation phases, online forecasting continuously adjusts the model during evaluation to adapt to potential concept drifts. Studies (Liu et al., 2016; Anava et al., 2013) extend statistical models (e.g., ARMA, ARIMA) to better capture dynamic temporal dependencies. Recently, online deep learning models have been proposed to further capture complex patterns within time series data. FSNet (Pham et al., 2022) introduces calibration module to dynamically balance fast adaptation to recent changes with the retention of prior knowledge. OneNet (Wen et al., 2023) incorporates reinforcement learning to model cross-variable and cross-time concept drifts. Addressing the information leakage issue in previous research, DSOF (Lau et al., 2025) redefines the OTSF setting and proposes a dual-stream mechanism to update model parameters. Nevertheless, prior studies do not explicitly model the patterns in input signals, hindering their ability to adapt to unobserved distributions. Additionally, their explicit storage of pattern information restricts the model's adaptability to extended online phases.

**Prompt-based Continual Learning.**   Rehearsal-free continual learning methods leverage the strong general representations of pre-trained models like ViT (Dosovitskiy et al., 2020). By fine-tuning only small, learnable "prompts" for each task, these methods achieve significant memory and computational efficiency, as the core model parameters remain unchanged. Rehearsal-free prompting strategies are divided into universal and specific methods. Universal approaches use a single set of prompts for all tasks. This includes methods like VPT (Jia et al., 2022), which optimizes a single prompt, L2P (Wang et al., 2022c), which uses a shared pool of prompts. In contrast, specific prompting methods, such as S-Prompts (Wang et al., 2022a), train a unique prompt for each individual task to address catastrophic forgetting. A hybrid method, DP (Wang et al., 2022b), employs both universal and specific prompts for all tasks simultaneously. However, the application of prompt learning to address distribution shifts in the time series domain remains unexplored. Given the continuous nature of time series data, the online learning scenario is more suitable than continual learning, which assumes distinct tasks. LLM4OT is the first to achieve the efficient and scalable prompting crucial for online learning scenarios by encoding knowledge from the underlying frequency basis.

**Frequency Analysis in Time Series Forecasting.**   Since time series data inherently exhibit complex temporal variations, frequency analysis techniques are employed to capture recurring patterns. Specifically, Discrete Fourier Transform (DFT) and Discrete Wavelet Transform (DWT) are commonly used to analyze the frequency content of time series data, with DFT focusing on global frequency components and DWT providing localized frequency information at different scales. FEDformer (Zhou et al., 2022b) proposes two possible structures to model signals in frequency domain, Fourier Enhanced Structure and Wavelet Enhanced Structure, each leveraging DFT and DWT, respectively. TimesNet (Wu et al., 2022) adopts Fourier Transform (FT) to capture temporal variations from multiple components with different period lengths. TimeMixer++ (Wang et al., 2024a) also leverages FT to capture periodic patterns from multi-scaled time points. STWave (Fang et al., 2021) and WaST (Nie et al., 2024) apply DWT to decompose the given signal into low and high frequency components to accurately capture spatio-temporal dependencies. Although prior studies effectively model dependencies in frequency domain, their approaches that utilize only the coefficient values without learning the underlying knowledge of each basis limits robustness to distribution shifts. On the other hand, LLM4OT can express unobserved pattern by a combination of basis embeddings by representing time series patterns as a combination of learnable frequency bases. We argue that this enhances the model's adaptability to distribution shifts, making it more suitable in online scenarios.

**Time Series Forecasting with Large Language Models.**   Recent advancements in Large Language Models (LLMs) have prompted researchers to investigate their transferability across a range of domains, including computer vision (CV) (Kim et al., 2024a; Liu et al., 2023a; 2024), recommendation systems (Kim et al., 2024b; Bao et al., 2023; Sanner et al., 2023), and Graph Neural Networks (GNN) (Chen et al., 2024; Ren et al., 2024). LLMs have also emerged as a promising tool for forecasting in data-sparse time series domains. LLM4TS (Chang et al., 2023) introduces two-stage fine-tuning approach to leverage LLMs for time series forecasting. GPT4TS (Zhou et al., 2023) retrains the positional embeddings and normalization layers of LLMs to preserve pre-trained knowledge while enhancing performance on downstream tasks. Additionally, TimeLLM (Jin et al., 2023) employs reprogramming method to align time series data with word embeddings. Similarly, TEST (Sun et al., 2023) and CALF (Liu et al., 2025) aim to align the two modalities to enable effective forecasting.

Building on the highlighted adaptability of LLMs in time series forecasting, we leverage these models to facilitate rapid adjustments in online scenarios.

## B  DATASET DETAILS

We evaluate our method on four wiedely used time sereis forecasting benchmarks.

**ETT.**   The ETT dataset[9] contains electricity load and oil temperature data collected from two electric power sites at 15-minute and hourly intervals over a span of two years. It includes two subsets: **ETTm1** and **ETTm2** (15-minute level) and **ETTh1** and **ETTh2** (hourly level), each with 6 covariates.

**Weather.**   The Weather dataset[10] comprises hourly meteorological observations collected from 21 monitoring stations distributed across various locations in the United States. Each station records a rich set of climate-related variables (i.e., 11 climate features recorded at nearly 1,600 locations across the U.S. at hourly intervals from 2010 to 2013), including but not limited to air temperature, dew point, relative humidity, atmospheric pressure, wind direction and speed, visibility, and precipitation amount.

**ECL.**   The ECL dataset[11] consists of electricity consumption data collected at 30-minute intervals from 321 residential and industrial clients in Portugal over a period of two years (from 2012 to 2014). Each client is represented as a distinct time series, capturing their individual electricity usage patterns.

**Traffic.**   The Traffic dataset[12] comprises occupancy rate measurements collected from 862 loop sensors installed on freeways in the San Francisco Bay Area. The data are recorded at a 5-minute sampling interval, offering fine-grained temporal resolution over an extended period. Each sensor captures the proportion of time a segment of road is occupied by vehicles, serving as a proxy for traffic density and congestion levels.

## C  BASELINE DETAILS

**DLinear (Zeng et al., 2023)**   DLinear simplifies the Transformer architecture by removing components such as positional encoding and global self-attention, based on the observation that the temporal order can be implicitly captured through the structure of the input patches. Instead, DLinear employs patch-wise tokenization and channel-independent processing to reduce complexity and improve generalization.

**PatchTST (Nie et al., 2022)**   PatchTST divides time series into patch-based input tokens for the Transformer, preserving local temporal patterns, reducing attention-related computational and memory costs, and enabling the use of longer input sequences. This design enhances long-term forecasting accuracy compared to other Transformer-based models. Additionally, PatchTST demonstrates strong performance in self-supervised pretraining and transfer learning settings.

**iTransformer (Liu et al., 2023b)**   iTransformer applies attention along the feature dimension rather than the temporal axis, capturing temporal dependencies with lightweight convolutions. By focusing attention on cross-variable interactions, iTransformer reduces computational cost while achieving strong performance on multivariate time series forecasting.

**TimeMixer (Wang et al., 2024b)**   TimeMixer proposes a novel multiscale-mixing paradigm, based on the observation that time series present distinct patterns in different sampling scales (fine and coarse). It is designed as a fully MLP-based architecture utilizing two key components: the Past-Decomposable-Mixing (PDM) block for history extraction and the Future-Multipredictor-Mixing (FMM) block for prediction. PDM applies decomposition to multiscale series, mixing seasonal and trend components separately to disentangle complex variations. This approach achieves consistent

---

[9]https://github.com/zhouhaoyi/ETDataset
[10]https://www.ncei.noaa.gov/data/local-climatological-data/
[11]https://archive.ics.uci.edu/ml/datasets/ElectricityLoadDiagrams20112014
[12]https://pems.dot.ca.gov/

strong performance in both long-term and short-term forecasting tasks with favorable run-time efficiency, capitalizing on its fully MLP-based design.

**LLM4TS (Chang et al., 2023)**  LLM4TS introduces a two-stage fine-tuning strategy that first aligns the LLM with the characteristics of time series data before fine-tuning it for specific forecasting tasks. To enhance the model's ability to capture complex temporal patterns, it incorporates a novel two-level aggregation method for integrating multi-scale information.

**GPT4TS (Zhou et al., 2023)**  GPT4TS adapts pre-trained language or vision models for time series analysis by freezing the core self-attention and feedforward layers. This approach, known as Frozen Pretrained Transformer (FPT), leverages the powerful representations from models trained on massive datasets to achieve strong performance across various time series tasks without task-specific architectural changes.

**Time-LLM (Jin et al., 2023)**  Time-LLM repurposes frozen LLMs for time series forecasting by first reprogramming the input time series into text prototypes to align it with the language modality. It then employs a Prompt-as-Prefix (PaP) strategy to enrich the input context, guiding the LLM to effectively reason over the reprogrammed time series patches for forecasting.

**FSNet (Pham et al., 2022)**  FSNet proposes a dual-path architecture that captures both short-term and long-term patterns via fast and slow learners. The fast learner quickly adapts to recent changes, while the slow learner models stable, long-term trends.

**OneNet (Wen et al., 2023)**  OneNet introduces an online ensemble framework. The key idea is to maintain multiple specialized models and adaptively weight them based on their recent performance, allowing the ensemble to respond effectively to non-stationary time series data.

**DSOF (Lau et al., 2025)**  DSOF redefines the OTSF setting such that the information leakage during model update is eliminated. It introduces a dual-stream framework that updates model parameters through distinct short- and long-term temporal context. The fast stream captures recent dynamics using short-term sequences, while the slow stream models stable patterns over longer horizons.

## D  IMPLEMENTATION DETAILS

In this section, we provide implementation details of LLM4OT.

### D.1  TRAINING DETAILS

**Model Training.**  Our method is implemented on Python 2.11, and Torch 2.2.2. In all our experiments, we use the AdamW optimizer for model optimization. The align module of Time-LLM (Jin et al., 2023) is used as the default align module. We train the model for 10 epochs during the training phase and perform one-step updates per data instance during the online phase to fine-tune the model. All experiments are conducted on a 48GB NVIDIA RTX A6000.

**Hyperparameters.**  The detailed hyperparameters used for model training are as follows: $\gamma = 0.3$ (in Eq. 5) for low-pass filtering, the level parameter $j = 2$ (in Eq. 7) for the discrete wavelet transform, and $\delta = 0.8$ (in Eq. 10) which assigns stronger supervision to near-future values.

## E  ALIGN MODULE

In this section, we introduce various modules for aligning time series with LLMs. We include alignment modules proposed in prior works (i.e., Time-LLM (Jin et al., 2023) and LLM4TS (Zhou et al., 2023)). These modules are designed to align time series with the language modality on which LLMs are trained.

### E.1  EXPLANATION OF THE OPERATION OF VARIOUS ALIGN MODULES

**Time-LLM (Jin et al., 2023)**  Time-LLM proposes reprogramming given patched time series representation $b(\mathbf{X}) = \hat{\mathbf{X}}_P \in \mathbb{R}^{P \times d}$, where $P$ denotes the number of input patches and $d$ denotes the

dimension of representation that time series backbone (i.e., $b(\cdot)$) generates, using pre-trained word embeddings $\mathbf{E} \in \mathbb{R}^{V \times D}$ from the backbone LLM, where $D$ is the hidden dimension of the backbone LLM. However, there is no prior information identifying which source tokens are directly relevant, leading to a large and potentially dense reprogramming space where using $\mathbf{E}$ alone. To address this, a compact set of text prototypes are constructed, denoted as $\mathbf{E}' \in \mathbb{E}^{V' \times D}$, where $V' < V$, by linearly probing $\mathbf{E}$. These prototypes capture essential language cues and are combined to represent local patch information while remaining within the language model's pre-trained space. To facilitate this interaction, a multi-head cross-attention layer is employed. Each head $k = \{1, \cdots, K\}$ uses query matrices $\mathbf{Q}_k = \hat{\mathbf{X}}_P \mathbf{W}_k^Q$, key matrices $\mathbf{K}_k = \mathbf{E}' \mathbf{W}_k^K$, and value matrices $\mathbf{V}_k = \mathbf{E}' \mathbf{W}_k^V$, where $\mathbf{W}_k^Q \in \mathbb{R}^{d \times d}$ and $\mathbf{W}_k^K, \mathbf{W}_k^V \in \mathbb{R}^{D \times d}$. The operation to reprogram time series patches in each attention head is defined as:

$$\mathbf{Z}_k = \text{ATTENTION}(\mathbf{Q}_k, \mathbf{K}_k, \mathbf{V}_k). \tag{11}$$

By aggregating the outputs $\mathbf{Z}_k$ from all attention heads, i.e., $\mathbf{Z}$, which is subsequently passed through a linear projection to align the hidden dimensions with those of the backbone model, resulting in the final output $\mathbf{X}^{align}$. Specifically, we align the patch-wise representations computed by the time-series backbone (i.e., $b(\mathbf{X}) = \hat{\mathbf{X}}_P$) with the language modality using the align module of the Time-LLM (i.e., $\mathbf{X}^{align}$), as illustrated in the architecture shown in Figure 2. During the training phase, $\mathbf{W}^Q$, $\mathbf{W}^K$, and $\mathbf{W}^V$ of the cross-attention layer are trained, whereas in online phase, these parameters are frozen.

**LLM4TS (Zhou et al., 2023)** To align pre-trained LLMs with the characteristics of time series data, LLM4TS treats the embeddings generated by the time series backbone (i.e., $b(\cdot)$) as token embeddings for each patch, i.e., $b(\mathbf{X}) = \hat{\mathbf{X}}_P = e_{\text{token}} \in \mathbb{R}^{P \times d}$, where $P$ denotes the number of patches and $d$ denotes the dimension of representation that time series backbone generates. To make these patch tokens compatible with the LLM's embedding space, LLM4TS introduces two additional types of embeddings. First, a positional embedding (i.e., $e_{\text{pos}}$) is added to provide the model with information about the order of patches, using a learnable lookup table $E_{\text{pos}}$:

$$e_{\text{pos}} = E_{\text{pos}}(i), \quad i = 1, ..., P. \tag{12}$$

Second, to incorporate multi-scale temporal cues such as hour, weekday, or holiday, LLM4TS introduces a temporal embedding constructed through two-level aggregation. Each temporal attribute is first embedded and summed to create a representation for each timestamp, and a pooling operation, typically by selecting the first timestamp, is then applied to produce a single temporal embedding for each patch:

$$e_{\text{temp}} = \text{Pooling}\Big( \sum_{a \in \{\text{sec}, \text{min}, \cdots\}} E_a(t_a) \Big), \tag{13}$$

where $a$ represents different temporal attributes (seconds, minutes, hours, holidays, etc.), $E_a$ denotes the trainable lookup table for each temporal attribute, $t_a$ are the series of patches containing temporal information for that temporal attribute, and Pooling applies the pooling method to the aggregated embeddings. The overall aligned embedding is then constructed by summing the three components:

$$\mathbf{X}^{align} = e_{\text{token}} + e_{\text{pos}} + e_{\text{temp}}. \tag{14}$$

In summary, we leverage the align module of the LLM4TS to align the patch-wise representations computed by the time-series backbone (i.e., $b(\mathbf{X}) = \hat{\mathbf{X}}_P$) with the language modality (i.e., $\mathbf{X}^{align}$), as illustrated in the architecture shown in Figure 2. During the training phase, $E_{pos}$ and $E_a$ are trained, then, during the online phase, these parameters are frozen.

### E.2 EXPERIMENTAL RESULTS ON VARIOUS ALIGN MODULES

In Table 5, we evaluate the performance of LLM4OT on the ETTh2 dataset using various align modules. Specifically, we adopt align modules from prior approaches (i.e., Time-LLM (Jin et al., 2023) and LLM4TS (Zhou et al., 2023)), which align time series with the language modality in static

Table 5: MSE and MAE results for prediction horizons of 1, 24, and 48 on ETTh2 data using various align modules.

| | | LLM4OT-Time-LLM | | LLM4OT-LLM4TS | |
|---|---|---|---|---|---|
| | $H$ | MSE | MAE | MSE | MAE |
| ETTh2 | 1 | 0.398 | 0.423 | 0.442 | 0.487 |
| | 24 | 0.932 | 0.607 | 1.089 | 0.598 |
| | 48 | 1.245 | 0.699 | 1.249 | 0.689 |

time series forecasting scenario. Since LLMs demonstrate strong transferability that supports effective adaptation in data-scarce online scenarios (i.e., Fig.1 (c)), aligning time series data into a form that the LLM can interpret allows it to be seamlessly combined with LLM4OT's pattern embedding and text description. Regardless of the prediction horizon, LLM4OT consistently outperforms the baselines shown in Table 1 across different align modules, demonstrating its robustness and align module-agnostic applicability.

Table 6: Standard deviations corresponding to the MSE results in Table 1.

| | $H$ | DLinear | PatchTST | iTransformer | TimeMixer | LLM4TS | GPT4TS | Time-LLM | FSNet | OneNet | DSOF | LLM4OT |
|---|---|---|---|---|---|---|---|---|---|---|---|---|
| ETTh1 | 1 | 0.022 | 0.036 | 0.043 | 0.021 | 0.231 | 0.311 | 0.175 | 0.489 | 0.124 | 0.012 | 0.011 |
| | 24 | 1.028 | 1.473 | 0.087 | 1.011 | 0.829 | 1.382 | 0.939 | 1.499 | 0.422 | 0.488 | 0.031 |
| | 48 | 0.917 | 1.429 | 0.919 | 0.978 | 0.947 | 1.422 | 0.994 | 1.926 | 0.553 | 0.913 | 0.035 |
| ETTh2 | 1 | 0.022 | 0.036 | 0.043 | 0.021 | 0.035 | 0.077 | 0.083 | 0.489 | 0.124 | 0.012 | 0.011 |
| | 24 | 1.028 | 1.473 | 0.087 | 1.074 | 0.327 | 0.249 | 0.531 | 1.499 | 0.422 | 0.488 | 0.031 |
| | 48 | 0.917 | 1.429 | 0.919 | 0.982 | 0.430 | 0.729 | 0.249 | 1.926 | 0.553 | 0.913 | 0.035 |
| ETTm1 | 1 | 0.012 | 0.014 | 0.014 | 0.012 | 0.015 | 0.011 | 0.010 | 0.012 | 0.021 | 0.011 | 0.002 |
| | 24 | 0.015 | 0.511 | 0.481 | 0.017 | 0.211 | 0.244 | 0.159 | 0.331 | 0.018 | 0.012 | 0.003 |
| | 48 | 0.021 | 0.907 | 0.449 | 0.020 | 0.221 | 0.253 | 0.183 | 0.251 | 0.122 | 0.015 | 0.001 |
| ETTm2 | 1 | 0.012 | 0.014 | 0.014 | 0.015 | 0.012 | 0.017 | 0.022 | 0.012 | 0.021 | 0.011 | 0.002 |
| | 24 | 0.015 | 0.511 | 0.481 | 0.016 | 0.233 | 0.342 | 0.337 | 0.331 | 0.018 | 0.012 | 0.003 |
| | 48 | 0.021 | 0.907 | 0.449 | 0.022 | 0.384 | 0.398 | 0.437 | 0.251 | 0.122 | 0.015 | 0.001 |
| WTH | 1 | 0.015 | 0.024 | 0.013 | 0.017 | 0.018 | 0.027 | 0.031 | 0.041 | 0.035 | 0.011 | 0.001 |
| | 24 | 0.344 | 0.412 | 0.455 | 0.302 | 0.369 | 0.312 | 0.424 | 0.772 | 0.022 | 0.013 | 0.001 |
| | 48 | 0.471 | 0.443 | 0.427 | 0.402 | 0.411 | 0.452 | 0.535 | 0.714 | 0.178 | 0.013 | 0.010 |
| ECL | 1 | 1.214 | 1.035 | 0.444 | 0.836 | 0.683 | 0.621 | 0.683 | 5.151 | 2.238 | 1.021 | 0.002 |
| | 24 | 2.221 | 2.011 | 1.788 | 1.883 | 1.429 | 1.588 | 1.524 | 6.138 | 4.192 | 0.813 | 0.003 |
| | 48 | 2.879 | 1.948 | 1.709 | 2.010 | 2.011 | 2.092 | 1.948 | 6.392 | 3.498 | 0.997 | 0.003 |
| Traffic | 1 | 0.018 | 0.018 | 0.019 | 0.015 | 0.011 | 0.018 | 0.020 | 0.041 | 0.022 | 0.015 | 0.003 |
| | 24 | 0.044 | 0.011 | 0.021 | 0.028 | 0.038 | 0.044 | 0.042 | 0.033 | 0.015 | 0.012 | 0.005 |
| | 48 | 0.121 | 0.029 | 0.021 | 0.103 | 0.055 | 0.048 | 0.049 | 0.067 | 0.017 | 0.013 | 0.011 |

# F ADDITIONAL RESULTS

## F.1 STANDARD DEVIATIONS

Table 6 presents the standard deviations corresponding to the experiments reported in Table 1. The standard deviations from three independent runs are provided for each dataset and model. LLM4OT exhibits more stable variation compared to the baselines, indicating its robust performance in data-scarce online scenarios thanks to the frequency-based pattern embedding's ability to adapt to diverse patterns, combined with the strong transferability of the LLM with the data augmentation provided by text descriptions.

## F.2 ONLINE UPDATING COST

This section compares the number of parameters updated during the online phase and the runtime statistics of online time series forecasting models to analyze their efficiency.

Table 7: The number of parameters updated during the online phase of OTSF models for prediction horizon 1 on the ETTh2 dataset.

| FSNet | OneNet | DSOF | LLM4OT |
|---|---|---|---|
| 2,037,115 | 1,018,045 | 1,236,349 | 897 |

**Parameters.** Table 7 reports the number of parameters updated during the online phase (i.e., those that are not frozen) for each OTSF model in experiments using the ETTh2 dataset. LLM4OT updates

significantly fewer parameters compared to the baselines, indicating its efficiency. This demonstrates that LLM4OT can leverage model adaptability to achieve strong performance (see Table 1) with fewer updated parameters. Updating many parameters during the online phase can make models vulnerable to distribution shifts and cause them to forget the base knowledge learned during training phase. In this regard, LLM4OT effectively and efficiently adapts to new data without forgetting previously learned knowledge by updating only a small subset of parameters.

Table 8: Comparison of runtime statistics between LLM4OT and existing OTSF methods for the scenario with a prediction horizon of 1 on each dataset, utilizing the total Training Phase Duration (sec), the total Online Phase Duration (sec), the Inference Latency (sec/itr), defined as the time required per update, and the GPU Memory Consumption (MiB) as metrics.

|  | Metric | ETTh1 | ETTh2 | ETTm1 | ETTm2 | WTH | ECL | Traffic |
|---|---|---|---|---|---|---|---|---|
| FSNet | Training Phase Duration (sec) | 275 | 274 | 801 | 830 | 775 | 398 | 159 |
| | Online Phase Duration (sec) | 341 | 332 | 1,024 | 1,019 | 998 | 463 | 351 |
| | Inference Latency (sec/itr) | 0.031 | 0.030 | 0.022 | 0.022 | 0.025 | 0.023 | 0.026 |
| | GPU Memory Consumption (MiB) | 379 | 386 | 377 | 384 | 392 | 390 | 385 |
| OneNet | Training Phase Duration (sec) | 551 | 540 | 1,621 | 1,633 | 1,503 | 848 | 303 |
| | Online Phase Duration (sec) | 701 | 690 | 2,237 | 2,241 | 2,079 | 994 | 741 |
| | Inference Latency (sec/itr) | 0.063 | 0.061 | 0.050 | 0.050 | 0.051 | 0.050 | 0.053 |
| | GPU Memory Consumption (MiB) | 461 | 451 | 452 | 447 | 439 | 448 | 452 |
| DSOF | Training Phase Duration (sec) | 622 | 613 | 1,904 | 1,936 | 1,789 | 994 | 379 |
| | Online Phase Duration (sec) | 733 | 705 | 2,588 | 2,559 | 2,371 | 1,201 | 855 |
| | Inference Latency (sec/itr) | 0.064 | 0.062 | 0.058 | 0.056 | 0.057 | 0.059 | 0.062 |
| | GPU Memory Consumption (MiB) | 528 | 519 | 538 | 575 | 583 | 561 | 565 |
| LLM4OT | Training Phase Duration (sec) | 4,650 | 4,650 | 14,428 | 14,501 | 13,786 | 7,249 | 2,788 |
| | Online Phase Duration (sec) | 787 | 741 | 2,938 | 2,973 | 2,841 | 1,389 | 931 |
| | Inference Latency (sec/itr) | 0.068 | 0.065 | 0.064 | 0.066 | 0.068 | 0.065 | 0.067 |
| | GPU Memory Consumption (MiB) | 42,331 | 42,319 | 42,906 | 43,008 | 43,213 | 42,839 | 43,012 |

**Runtime Comparison.** Table 8 presents the runtime statistics for each dataset in the scenario where the prediction horizon is 1. We utilize the total Training Phase Duration (sec), the total Online Phase Duration (sec), the Inference Latency (sec/itr), defined as the time required per update, and the GPU Memory Consumption (MiB) as metrics. OTSF requires a model to be sufficiently pre-trained on initial data (i.e., training phase) and then continually adapted to the subsequent data stream (i.e., online phase). Consequently, resource optimization must prioritize the cost-sensitive online phase over the training phase. The duration of the online phase and the inference latency for LLM4OT require only slightly more time compared to existing OTSF models. This demonstrates that LLM4OT continuously adapts to new online distributions with significantly less parameter tuning (see Table 7) by simultaneously leveraging the LLM's superior transferability and the distribution guidance from the frequency basis-based prompting method. While LLM4OT incurs a higher initial cost for training duration and GPU memory compared to existing models, this cost is justifiable as the training phase is less resource-sensitive and prioritizes the sufficient acquisition of base knowledge. Furthermore, considering the current prevalence of LLM-based time series models and time series foundation models, the memory consumption is not unduly burdensome. In summary, LLM4OT achieves significantly superior performance (please refer to Table 1) in the cost-sensitive online phase with only a comparable cost, despite its larger initial overhead during the initial training.

Table 9: Performance comparison of LLM4OT using various backbone LLMs on the ETTh2 dataset for prediction horizons of 1, 24, and 48.

|  |  | LLM4OT-OPT(1.3B) | | LLM4OT-Llama(3B) | | LLM4OT-Llama(7B) | | LLM4OT-Llama(13B) | |
|---|---|---|---|---|---|---|---|---|---|
|  | $H$ | MSE | MAE | MSE | MAE | MSE | MAE | MSE | MAE |
| ETTh2 | 1 | 0.435 | 0.436 | 0.403 | 0.434 | 0.398 | 0.423 | 0.401 | 0.428 |
| | 24 | 0.977 | 0.623 | 0.938 | 0.608 | 0.932 | 0.607 | 0.931 | 0.611 |
| | 48 | 1.266 | 0.702 | 1.259 | 0.705 | 1.245 | 0.699 | 1.231 | 0.702 |

## F.3 MODEL PERFORMANCE USING VARIOUS LLMS

Table 9 compares the MSE and MAE performance of LLM4OT using various backbone LLMs of varying sizes (i.e., Llama-3B, Llama-7B, Llama-13B, and OPT-1.3B) for forecasting prediction horizons of 1, 24, and 48 on the ETTh2 dataset. The results demonstrate a clear trend where performance

improves as the size of the backbone LLM increases. This improvement can be attributed to the larger capacity of these models, which encodes more comprehensive knowledge, indicating that the knowledge embedded in LLMs benefits the OTSF task. However, it is important to note that even when utilizing smaller models such as OPT-1.3B and Llama-3B as the backbone, LLM4OT consistently outperforms the baseline performance reported in Table 1. This demonstrates that LLM4OT is a backbone LLM-agnostic framework and highlights the effectiveness of LLMs' strong transferability in facilitating model adaptation in data-scarce online scenario. Moreover, the synergy between the frequency basis-based pattern embedding, which captures overall patterns, and the text description, which enriches the data by providing recent pattern information in textual form, further enhances the transferability of the backbone LLM regardless of its type, resulting in robust performance in OTSF.

Table 10: Comparison of MSE and MAE results across various datasets under scenarios with an shortened online phase, where the train/valid/test split is set to 20%/5%/30%.

|  |  | ETTh2 | | | ETTm1 | | | WTH | | | ECL | | | Traffic | | |
|---|---|---|---|---|---|---|---|---|---|---|---|---|---|---|---|---|
|  |  | 1 | 24 | 48 | 1 | 24 | 48 | 1 | 24 | 48 | 1 | 24 | 48 | 1 | 24 | 48 |
| FSNet | MSE | 6.134 | 13.23 | 14.98 | 0.132 | 1.038 | 1.654 | 0.534 | 1.421 | 2.011 | 178 | 329 | 386 | 0.473 | 0.508 | 0.638 |
|  | MAE | 2.376 | 3.537 | 3.774 | 0.333 | 0.988 | 1.186 | 0.707 | 1.092 | 1.358 | 12.43 | 16.85 | 16.99 | 0.587 | 0.674 | 0.728 |
| OneNet | MSE | 1.946 | 5.021 | 6.664 | 0.104 | 1.004 | 1.084 | 0.397 | 1.198 | 2.307 | 19.33 | 66.37 | 107.39 | 0.188 | 0.399 | 0.538 |
|  | MAE | 1.294 | 2.047 | 2.381 | 0.222 | 0.992 | 0.945 | 0.538 | 1.004 | 1.418 | 4.096 | 7.467 | 9.062 | 0.339 | 0.532 | 0.693 |
| DSOF | MSE | 0.448 | 2.001 | 3.215 | 0.084 | 0.402 | 0.425 | 0.272 | 0.558 | 0.757 | 3.524 | 3.771 | 4.211 | 0.203 | 0.298 | **0.304** |
|  | MAE | **0.569** | 1.314 | 1.693 | 0.249 | **0.534** | 0.632 | 0.491 | 0.696 | 0.827 | 1.727 | 1.901 | 2.007 | **0.406** | 0.515 | **0.526** |
| LLM4OT | MSE | **0.372** | **0.883** | **1.027** | **0.071** | **0.302** | **0.389** | **0.047** | **0.102** | **0.206** | **0.147** | **0.254** | **0.333** | 0.201 | **0.275** | 0.313 |
|  | MAE | 0.599 | **0.899** | **0.913** | **0.244** | 0.539 | **0.603** | **0.196** | **0.299** | **0.418** | **0.324** | **0.483** | **0.524** | 0.423 | **0.504** | 0.539 |

## F.4 ALTERNATIVE SCENARIO FOR EXTENDED ONLINE PHASE ANALYSIS

In Table 10, to fairly evaluate how LLM4OT and existing online time series forecasting methods perform as the online phase becomes longer, we conducted a comparative analysis on five datasets where the duration of the training phase remains the same as our default setting (with a 20%/5%/75% train/valid/test split), but the test set is shortened to 30%, resulting in a 20%/5%/30% split. This enables a direct comparison of performance degradation under a longer online phase. By comparing with the results in Table 1, we observe that LLM4OT exhibits a significantly smaller margin of performance degradation as the online phase extends from 30% to 75% of the data, compared to existing time series forecasting methods (i.e., FSNet, OneNet, and DSOF). This suggests that the excellent transferability of the LLM aligns well with the time series backbone's ability to understand time series data, allowing it to maintain superior performance even in prolonged, data-scarce online scenarios. Furthermore, the frequency basis-driven pattern embeddings and text descriptions provide both overall and recently specific guidance on the data distribution, enabling robust and continuous adaptation to the continuous distribution shifts that occur during the online phase.

Table 11: Average cosine similarity between prompts in the prompt bank, generated by three strategies, for the prediction horizon1 in the ETTh2 dataset.

|  | $\mathbf{P}_{low}^2$ | $\mathbf{P}_{low}^3$ |
|---|---|---|
| Similarity with $\mathbf{P}_{low}^1$ | 0.7375 | -0.1827 |

## F.5 GENERALIZABILITY OF FREQUENCY BASIS-BASED PROMPTING STRATEGY

In Table 11, to demonstrate that the combination of knowledge encoded from each frequency basis can effectively represent unseen patterns in the online phase, even with learning solely from the training phase, we compare three Prompt Banks (i.e., $\mathbf{P}_{low}^1$, $\mathbf{P}_{low}^2$, and $\mathbf{P}_{low}^3$) generated using the following strategies:

- $\mathbf{P}_{low}^1$ (Optimal): The prompt bank is trained during the training phase and further trained during the online phase.

- $\mathbf{P}_{low}^2$ (LLM4OT): The prompt bank is trained during the training phase, and frozen during the online phase.

- $\mathbf{P}_{low}^3$: The prompt bank is randomly generated.

The cosine similarity between the optimal Prompt Bank 1 (i.e., $\mathbf{P}_{low}^1$) and each of Prompt Bank 2 (i.e., $\mathbf{P}_{low}^2$) and Prompt Bank 3 (i.e., $\mathbf{P}_{low}^3$) are shown in Table 11. $\mathbf{P}_{low}^1$ is the optimal prompt bank that can be obtained when unseen patterns (i.e., patterns in online phase) are included in the training. $\mathbf{P}_{low}^2$ shows a high degree of similarity to the $\mathbf{P}_{low}^1$, both in absolute terms and especially

when compared to $\mathbf{P}_{low}^3$. This indicates that, even without additional learning during the online phase, prompt learning based on frequency bases can effectively represent incoming new patterns and guide the model.

### F.6 ADDITIONAL ABLATION STUDIES

This section presents an ablation study to analyze the contributions of each component of LLM4OT in a more practical yet challenging cross-dataset (Sec. F.6.1), and extended online phase (Sec. F.6.2) scenarios.

Table 12: Ablation study of each component of the LLM4OT in a cross-dataset scenario. MSE results for each scenario are shown.

| | $\mathcal{P}$ | $\mathcal{T}$ | ETTh1 → ETTh2 | | | ETTh2 → ETTh1 | | | ETTm1 → ETTh2 | | | ETTm2 → ETTh2 | | |
|---|---|---|---|---|---|---|---|---|---|---|---|---|---|---|
| | | | 1 | 24 | 48 | 1 | 24 | 48 | 1 | 24 | 48 | 1 | 24 | 48 |
| (1) | ✗ | ✗ | 1.027 | 2.018 | 3.689 | 1.169 | 1.883 | 3.015 | 1.381 | 2.447 | 4.107 | 1.268 | 2.317 | 3.116 |
| (2) | ✓ | ✗ | 0.645 | 1.448 | 1.935 | 0.459 | 1.266 | 1.957 | 0.730 | 1.545 | 2.004 | 0.739 | 1.371 | 1.997 |
| (3) | ✗ | ✓ | 0.879 | 1.709 | 2.879 | 0.829 | 1.670 | 2.731 | 0.992 | 1.718 | 2.614 | 1.002 | 1.923 | 2.499 |
| (4) | ✓ | ✓ | **0.427** | **1.122** | **1.413** | **0.399** | **0.892** | **1.277** | **0.521** | **1.229** | **1.418** | **0.513** | **1.038** | **1.535** |

### F.6.1 CROSS-DATASET SCENARIO

In Table 12, using the ETT series datasets, we conduct an ablation study of the results shown in Table 3, where we presented results across four cross-dataset scenarios. We measure the performance with and without pattern embedding (i.e., $\mathcal{P}$) and text description (i.e., $\mathcal{T}$), and make the following key observations: **(1)** Pattern embedding effectively helps address distribution shift (Row (1) vs. (2)). Considering the sequential nature of time series data and the characteristics of cross-dataset scenarios, significant distribution shifts occur, requiring the model to adapt appropriately during the online phase. Directly storing pattern information, as done in prior studies (i.e., FSNet (Pham et al., 2022), OneNet (Wen et al., 2023), and DSOF (Lau et al., 2025)), limits meaningful retrieval when encountering patterns dissimilar to those previously observed, thereby failing to handle distribution shifts. In contrast, representing time series patterns as combinations of frequency bases and learning the knowledge associated with each basis enables the model to generate reliable pattern embeddings for unseen patterns, allowing for more robust adaptation. **(2)** Text descriptions aid model adaptation in the presence of distribution shifts (Row (1) vs. (3)). In data-scarce online phases, especially when distribution shifts occur, text descriptions enrich the limited data. Furthermore, by providing information about recent shifted patterns from both time and frequency domain perspectives, they enable the model to recognize shifted distributions without additional training, resulting in improved adaptation ability. **(3)** Utilizing pattern embedding and text description together produces a synergistic effect (Row (2&3) vs. (4)). Pattern embedding captures the overall time series pattern to address distribution shifts, while text descriptions amplify information about recent shifted patterns, facilitating adaptation with richer data.

Table 13: Ablation study of each component of the LLM4OT under scenarios with an extended online phase, where the train/valid/test split is set to 10%/5%/85%. MSE results for each scenario are shown.

| | $\mathcal{P}$ | $\mathcal{T}$ | ETTh2 | | | ETTm1 | | | WTH | | | ECL | | | Traffic | | |
|---|---|---|---|---|---|---|---|---|---|---|---|---|---|---|---|---|---|
| | | | 1 | 24 | 48 | 1 | 24 | 48 | 1 | 24 | 48 | 1 | 24 | 48 | 1 | 24 | 48 |
| (1) | ✗ | ✗ | 1.099 | 2.008 | 4.118 | 0.102 | 0.633 | 0.777 | 0.379 | 0.663 | 1.081 | 2.886 | 2.969 | 3.131 | 0.931 | 1.312 | 1.444 |
| (2) | ✓ | ✗ | 0.681 | 1.339 | 1.803 | 0.092 | 0.498 | 0.551 | 0.147 | 0.284 | 0.398 | 1.039 | 1.231 | 1.334 | 0.499 | 0.473 | 0.505 |
| (3) | ✗ | ✓ | 0.758 | 1.504 | 2.033 | 0.094 | 0.531 | 0.589 | 0.192 | 0.379 | 0.522 | 1.479 | 1.483 | 1.611 | 0.612 | 0.566 | 0.772 |
| (4) | ✓ | ✓ | **0.412** | **0.978** | **1.305** | **0.087** | **0.411** | **0.487** | **0.051** | **0.103** | **0.223** | **0.179** | **0.291** | **0.362** | **0.485** | **0.356** | **0.379** |

### F.6.2 EXTENDED ONLINE PHASE SCENARIO

In Table 13, we perform an ablation study of the results reported in Table 4, where we presented the results under a scenario with an extended online phase. We measure the performance with and without pattern embedding (i.e., $\mathcal{P}$) and text description (i.e., $\mathcal{T}$), and make the following key observations: **(1)** Pattern embedding captures the overall time series pattern as the online phase progresses, preventing significant performance degradation due to distribution shifts (Row (1) vs. (2)). As the online phase progresses, a wider variety of patterns emerge, and unseen patterns occur

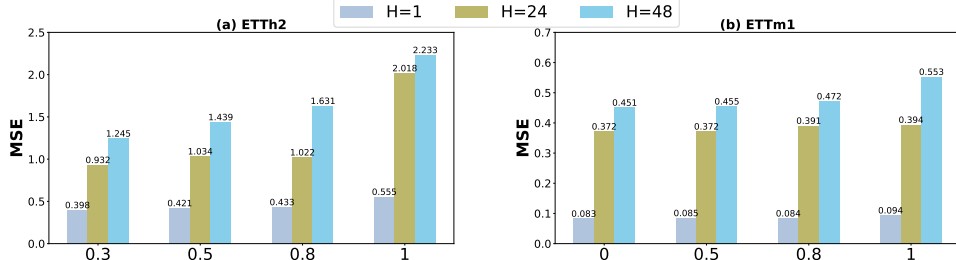

Figure 3: Sensitivity analysis for the hyperparameter $\gamma$ in Equation 5, which controls the low-pass filtering. (a) MSE results on the ETTh2 dataset for prediction horizons of 1, 24, and 48, as the $\gamma$ varied is varied 0.3, 0.5, 0.8, and 1. (b) MSE results on the ETTm1 dataset for prediction horizons of 1, 24, and 48, as the $\gamma$ varied is varied 0.3, 0.5, 0.8, and 1.

more frequently. By decomposing time series into underlying frequency bases and storing knowledge for each basis instead of directly storing patterns, the model can robustly provide accurate pattern information even in the presence of various and unseen patterns, by combining the knowledge of each basis to generate pattern embeddings. **(2)** Text descriptions mitigate the data scarcity problem that is exacerbated as the online phase progresses, preventing performance degradation (Row (1) vs. (3)). As the online phase extends, the amount of data available for the model to learn from significantly decreases, leading to performance degradation. However, text descriptions provide information about recent patterns to the model without any additional training, enabling the model to efficiently and effectively maintain its adaptation performance even in data-scarce scenarios. **(3)** The combination of pattern embedding and text description creates a strong synergy in the extended online phase scenario (Row (2&3) vs. (4)). The ability of pattern embedding to capture the overall time series pattern without being disrupted by distribution shifts, along with the ability of text descriptions to amplify information about recent patterns, maximizes the model's robustness as the online phase progresses. This demonstrates that LLM4OT can be effectively used in long online phases without frequent retraining, making it a practical model.

### F.7 SENSITIVITY ANALYSIS

In this section, we present a sensitivity analysis for the hyperparameters $\gamma$ (in Equation 5) and $\delta$ (in Equation 10) utilized in LLM4OT.

### F.7.1 HYPERPARAMETER $\gamma$

To analyze the sensitivity of LLM4OT to the hyperparameter $\gamma$, which is used to filter for significant low-frequency information when generating pattern embeddings, we conduct an experiment presented in Figure 3. Using the ETTh2 and ETTm1 datasets, we vary the $\gamma$ parameter in Equation 5 across values of 0.3, 0.5, 0.8, and 1, and observe the corresponding MSE for prediction horizons of 1, 24, and 48. On both datasets, optimal performance is achieved when $\gamma = 0.3$. Performance degrades as more high-frequency components are retained, with the most significant drop observed when $\gamma = 1$. This indicates that when generating pattern embeddings that contain knowledge from each frequency component to capture the overall distribution of the time series data, the high-frequency components are largely irrelevant to the overall distribution and instead act as noise.

### F.7.2 GEOMETRIC DECAY FACTOR $\delta$

To investigate the sensitivity of LLM4OT to the geometric decay factor $\delta$, which is utilized to reduce the influence of pseudo-label unreliability and prediction errors for timestamps distant from the current observation when the prediction horizon is greater than 1, we conduct an experiment presented in Figure 4. Using the ETTh2 and ETTm1 datasets, we vary the $\delta$ parameter in Equation 10 across values of 0.3, 0.5, 0.8, and 1 and observe the corresponding MSE for prediction horizons of 24 and 48. On both datasets, optimal performance is achieved when $\delta$ is around 0.8. A $\delta$ value of 1 which signifies no decay effect, leads to a performance drop. This is due to the netgative influence of unreliable pseudo-labels generated by the time series backbone and the less accurate predictions for distant timestamps. Therefore, to robustly adapt to continuous distribution shifts, we utilize an appropriate geometric decay factor to mitigate noise during the model's training process. We observed that a similar value of of $\delta$ (i.e., $\delta = 0.8$) is consistently effective across various datasets.

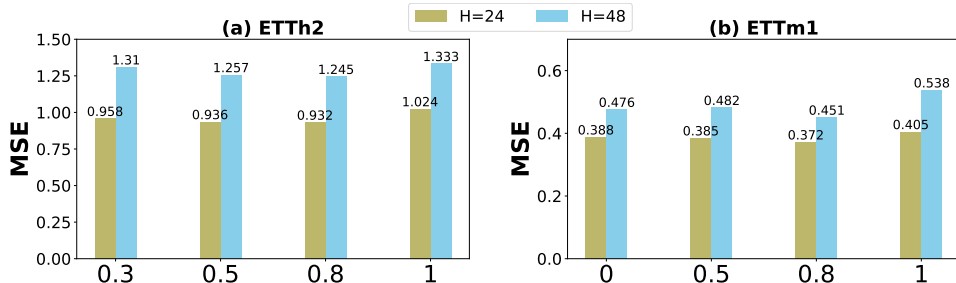

Figure 4: Sensitivity analysis of the geometric decay factor $\delta$ in Equation 10. (a) MSE results on the ETTh2 dataset for prediction horizons of 24, and 48, as the $\delta$ varied is varied 0.3, 0.5, 0.8, and 1. (b) MSE results on the ETTm1 dataset for prediction horizons of 24, and 48, as the $\delta$ varied is varied 0.3, 0.5, 0.8, and 1.

### F.8    MODEL PERFORMANCE WITH ADDITIONAL PARAMETER TUNING

In this section, we analyze the LLM4OT's performance and updating cost (i.e., the number of updated parameters and the duration of the online phase) when additional parameter tuning is performed during the online phase. In F.8.1, the time series backbone network (i.e., $b(\cdot)$) is tuned during the online phase, whereas in F.8.2, the prompt bank (i.e., $\mathbf{P}$) and the align module are tuned.

#### F.8.1    TIME SERIES BACKBONE NETWORK

In Table 14, expanding on our previous approach of tuning only the projector, we now also fine-tune the time series backbone network (i.e., $b(\cdot)$) and evaluate the impact on performance, the number of parameters updated during the online phase, and the total time required for the online phase on ETTh2 dataset. While fine-tuning the time-series backbone network slightly improves performance compared to the results in Table 1, it comes at the cost of a noticeable increase in parameter updates and a decrease in update speed. Therefore, to reduce the online update cost, LLM4OT  successfully minimizes tuning without a significant performance drop by guiding the distribution through pattern embeddings and text descriptions.

Table 14: Performance, number of updated parameters, and the total time (in seconds) required for the online phase when tuning the time series backbone network (i.e., $b(\cdot)$) in the online phase on the ETTh2 dataset for prediction horizons of 1, 24, and 48.

|  | $H$ | MSE | MAE | Updated Parameters | Online Time (sec) |
|---|---|---|---|---|---|
| ETTh2 | 1 | 0.385 | 0.604 | 3,073 | 847 |
|  | 24 | 0.911 | 0.925 | 23,704 | 958 |
|  | 48 | 1.159 | 0.976 | 45,232 | 993 |

#### F.8.2    PROMPT BANK AND ALIGN MODULE

In Table 15, we build upon our previous approach of tuning only the projector by additionally fine-tuning the prompt bank (i.e., $\mathbf{P}$) and the alignment module. It shows the resulting impact on performance, the number of updated parameters, and the total online phase time for the ETTh2 dataset. Similar to the results in Section F.8.1, a slight performance improvement is observed compared to the results in Table 1 as more components are tuned during the online phase. However, this comes with a loss in efficiency, as the number of updated parameters and the required time for the online phase increase noticeably. Since efficiency is a critical factor in online scenarios, LLM4OT achieves superior performance by only tuning the output projector, which minimizes the number of parameter updates. This is made possible by effectively guiding the distribution through pattern embeddings and text descriptions.

### F.9    MODEL PERFORMANCE UNDER SCENARIOS WITH INFORMATION LEAKAGE

Note that in Table 1 we reported results of OTSF without information leakage following DSOF (Lau et al., 2025), where we showed the superiority of LLM4OT compared with baselines. In Table 16, we

Table 15: Performance, number of updated parameters, and the total time (in seconds) required for the online phase when tuning the prompt bank (i.e., $\mathbf{P}$) and align module in the online phase on the ETTh2 dataset for prediction horizons of 1, 24, and 48.

| | $H$ | MSE | MAE | Updated Parameters | Online Time (sec) |
|---|---|---|---|---|---|
| ETTh2 | 1 | 0.381 | 0.597 | 1,269,441 | 1,020 |
| | 24 | 0.897 | 0.931 | 1,290,072 | 1,143 |
| | 48 | 1.168 | 1.007 | 1,293,660 | 1,402 |

Table 16: Comparison of the performance between prior methods (i.e., FSNet (Pham et al., 2022) and OneNet (Wen et al., 2023)) and LLM4OT with information leakage. Evaluation is conducted on five public benchmarks using MSE and MAE metrics for prediction horizons of 1, 24, and 48.

| | | ETTh2 | | | ETTm1 | | | WTH | | | ECL | | | Traffic | | |
|---|---|---|---|---|---|---|---|---|---|---|---|---|---|---|---|---|
| | | 1 | 24 | 48 | 1 | 24 | 48 | 1 | 24 | 48 | 1 | 24 | 48 | 1 | 24 | 48 |
| FSNet | MSE | 8.916 | 4.536 | 15.687 | 0.139 | 0.790 | 0.908 | 0.218 | 0.994 | 1.289 | 31.553 | 25.842 | 43.960 | 0.506 | 0.641 | 0.621 |
| | MAE | 0.994 | 0.940 | 1.741 | 0.225 | 0.594 | 0.643 | 0.423 | 0.887 | 1.021 | 0.598 | 0.786 | 0.928 | 0.323 | 0.390 | 0.384 |
| OneNet | MSE | 0.389 | 0.508 | 0.876 | 0.102 | 0.217 | **0.111** | 0.132 | 0.799 | 0.857 | 2.607 | 3.134 | 3.021 | **0.249** | 0.413 | 0.451 |
| | MAE | 0.353 | 0.403 | 0.522 | 0.184 | 0.310 | **0.223** | 0.357 | 0.824 | 0.851 | 0.253 | 0.393 | 0.378 | 0.207 | 0.296 | 0.318 |
| LLM4OT | MSE | **0.369** | **0.455** | **0.743** | **0.092** | **0.099** | 0.113 | **0.004** | **0.009** | **0.018** | **0.153** | **0.239** | **0.311** | 0.254 | **0.389** | **0.427** |
| | MAE | **0.313** | **0.384** | **0.498** | **0.182** | **0.298** | 0.241 | **0.027** | **0.060** | **0.093** | **0.222** | **0.343** | **0.371** | **0.121** | **0.262** | **0.311** |

compare the performance of FSNet (Pham et al., 2022), OneNet (Wen et al., 2023), and LLM4OT in a scenario *with information leakage*, as these approaches were originally designed under settings that allow information leakage. We observe that the performance of LLM4OT as well as FSNet and OneNet enhances under the presence of information leakage, while LLM4OT still demonstrating superior performance. This is primarily because, even though FSNet and OneNet can leverage more abundant information for updates in scenarios with information leakage, they become unreliable when a distribution shift occurs, as they rely on directly storing time series patterns and retrieving them based on similarity. In contrast, LLM4OT decomposes patterns into underlying frequency bases, learning knowledge for each basis. By decomposing patterns as combinations of these bases, LLM4OT can handle shifted patterns by generating pattern embeddings using combinations of learned frequency basis knowledge. Furthermore, LLM4OT leverages a pre-trained LLM alongside a time series backbone, combining the knowledge of the time series backbone model with the excellent transferability of the LLM. This helps the model adapt better to new data during the data-scarce online phase. Additionally, to boost the model's adaptability in data-scarce scenario, LLM4OT leverages text descriptions to provide recent pattern information from both time and frequency domains, effectively enriching the limited data. In conclusion, LLM4OT demonstrates superior performance compared to the baseline, regardless of the presence of information leakage.

## G  PSEUDOCODE

---

**Algorithm 1** Pseudocode for training and online phases of LLM4OT

---

1: **Input:** Time Series $\mathbf{X} \in \mathbb{R}^{N_{data}}$, Pre-trained LLM, Batch Size, Align Module
2: **Output:** Time Series Backbone Network $b(\cdot)$, Prompt Bank $\mathbf{P}$, Output Projection Layer
3:
4: **# Training Phase**
5: Freeze the Pre-trained LLM
6: **for** i in range($\frac{N_{train}}{\text{Batch Size}}$) **do**
7:    $\mathbf{X}^{(i)} \in \mathbb{R}^{\text{Batch Size} \times L}$                                 ▷ $i$-th batch for batch training
8:
9:    **# Aligning**
10:    $emb_{\mathbf{X}^{(i)}} = b(\mathbf{X}^{(i)})$
11:    Generate $emb_{\mathbf{X}^{(i)}}^{align}$ using the Align Module
12:
13:    **# Text Description**
14:    Generate $\mathcal{T}_{\mathbf{X}^{(i)}}$ using $text_{\mathbf{X}^{(i)}}$
15:
16:    **# Pattern Embedding**
17:    Decompose $\mathbf{X}^{(i)}$ into frequency bases and coefficients using the DFT         ▷ Eq. 4
18:    Generate $\mathcal{P}_{\mathbf{X}^{(i)}}$                                          ▷ Eq. 5
19:
20:    Concatenate $\mathcal{P}_{\mathbf{X}^{(i)}}$, $\mathcal{T}_{\mathbf{X}^{(i)}}$, and $emb_{\mathbf{X}^{(i)}}^{align}$ as input to the Pre-trained LLM
21:    The Pre-trained LLM's representation is projected to obtain $\hat{\mathbf{X}}^{(i)}$
22:
23:    Train all parameters except the Pre-trained LLM using $\mathcal{L}_{training}^{+}$          ▷ Eq. 9
24: **end for**
25:
26: **# Online Phase**
27: Freeze all parameters except those in the Output Projection Layer
28: **for** i in range($N_{online}$) **do**
29:    $\mathbf{X}^{(i)} \in \mathbb{R}^{L}$                                       ▷ $i$-th data instance
30:
31:    **# Aligning**
32:    $emb_{\mathbf{X}^{(i)}} = b(\mathbf{X}^{(i)})$
33:    Generate $emb_{\mathbf{X}^{(i)}}^{align}$ using the Align Module
34:
35:    **# Text Description**
36:    Generate $\mathcal{T}_{\mathbf{X}^{(i)}}$ using $text_{\mathbf{X}^{(i)}}$
37:
38:    **# Pattern Embedding**
39:    Decompose $\mathbf{X}^{(i)}$ into frequency bases and coefficients using the DFT         ▷ Eq. 4
40:    Generate $\mathcal{P}_{\mathbf{X}^{(i)}}$                                          ▷ Eq. 5
41:
42:    Concatenate $\mathcal{P}_{\mathbf{X}^{(i)}}$, $\mathcal{T}_{\mathbf{X}^{(i)}}$, and $emb_{\mathbf{X}^{(i)}}^{align}$ as input to the Pre-trained LLM
43:    The Pre-trained LLM's representation is projected to obtain $\hat{\mathbf{X}}^{(i)}$
44:
45:    Update the Output Projection Layer using $\mathcal{L}_{online}$                  ▷ Eq. 10
46: **end for**
47:
48: Evaluate the online phase                                             ▷ Eq. 1

