# OpenReview forum: "LLM-based Online Time Series Forecasting with Frequency-driven Pattern Recognition"
_ICLR.cc/2026/Conference — Submitted to ICLR 2026_

### Official Review · Reviewer_Sjmn · 2025-10-27

**Soundness:** 3
**Presentation:** 2
**Contribution:** 3
**Rating:** 4
**Confidence:** 4

**Summary:**

The paper introduces a novel LLM-based framework LLM4OT for online time series forecasting (OTSF) that integrates frequency-domain pattern learning and language-driven adaptation. The framework represents time series as combinations of frequency bases to capture underlying temporal patterns, incorporates a pre-trained LLM aligned with a time-series backbone network, and leverages text-based recent pattern descriptions to enhance adaptation under data-scarce online scenarios. The proposed framework is evaluated on multiple real-world datasets across different domains against three categories of baselines—static forecasting models, LLM-based models, and online forecasting methods. Experiments, including ablation studies, cross-dataset robustness tests, and extended online phase analyses, comprehensively demonstrate that LLM4OT achieves superior adaptability and robustness under continuous distribution shifts.

**Strengths:**

(a) The paper presents a novel formulation of online time series forecasting (OTSF) by integrating large language models (LLMs) with frequency-domain pattern recognition. The idea of decomposing time series into frequency bases and encoding these into prompts offers a creative way to represent evolving temporal dynamics beyond traditional buffer-based or dual-stream OTSF designs.

(b) The paper also introduces text-based recent pattern descriptions, allowing pre-trained LLMs to exploit semantic priors and adapt effectively to data-scarce online scenarios without additional training. The combination of frequency-based pattern embeddings, aligned time-series representations, and text-based recent pattern descriptions provides a new perspective on achieving adaptability under continuous distribution shifts.

(c) Extensive experiments across multiple real-world datasets (ETT, Weather, ECL, Traffic) and ablation studies (including cross-dataset and extended online-phase evaluations) support the proposed framework. The results demonstrate improvements over both traditional OTSF models (e.g., FSNet, OneNet, DSOF) and recent LLM-based time series baselines.

**Weaknesses:**

(a) In the framework description, the paper only briefly mentions that the pattern embedding, text embedding, and aligned time-series embedding are concatenated and then fed into the pre-trained LLM. However, it does not clearly explain how the text representation embedding is generated or whether any additional information (such as task prompts, positional encodings, or structural tokens) is included after concatenation. These details should have been clarified earlier in the framework section. The current description makes it difficult to understand how the inputs are constructed and how the LLM processes them.

(b) According to Appendix F.5, the prompt bank is trained jointly with the backbone and alignment module during the training phase and then frozen during the online phase. However, this design choice is only mentioned later in the appendix instead of being clearly described in the main framework section. It should have been explicitly introduced earlier to help readers understand how the prompts are constructed and integrated into the overall architecture.

(c) Although Appendix F.2 and Table 8 report online update times and the number of parameters updated, the analysis remains limited to a single dataset and focuses only on per-step update efficiency. The framework depends on large-scale LLMs (e.g., Llama-7B or 13B), but the paper provides no discussion of the overall computational cost, inference latency, or memory footprint. Without details on GPU utilization, runtime scalability, or hardware configurations, it is difficult to assess the true efficiency and deployability of LLM4OT in realistic online forecasting scenarios. A more comprehensive resource and scalability analysis would be necessary to substantiate the claimed efficiency.

**Questions:**

(a) Could the authors clarify how the text representation embedding is generated before being concatenated with the pattern and aligned time-series embeddings?

(b) When concatenating the three embeddings (pattern, text, and aligned time-series), is any normalization, positional encoding, or token-type distinction applied before feeding them into the LLM? If not, how does the model differentiate between heterogeneous sources of information?

(c) Regarding the prompt bank, could the authors explain on how initialization strategy was chosen, and whether the prompts share the same embedding space as the pre-trained LLM tokens? This information seems crucial for reproducibility.

(d) The paper claims that LLM4OT is efficient during the online phase, however, the reported analysis (Appendix F.2 and Table 8) only covers update speed on a single dataset. Could the authors provide more comprehensive runtime statistics such as total training time, GPU memory usage, or inference latency?

---

> ### Author Response · Authors · 2025-11-21
>
> **W.(a) & Q.(a) & Q.(b)**
>
> We appreciate the insightful suggestion. First, we will explain in detail how the embeddings for the text descriptions are generated. The process of embedding these text descriptions does not involve any training phase. To generate the text embeddings, the text description is first processed by the pre-trained LLM’s tokenizer. The resulting token IDs are then passed through the LLM’s frozen input embedding layer to retrieve their corresponding token embeddings. This sequence of text token embeddings, which we refer to as the text embedding, is concatenated with the pattern embedding and the aligned time series embedding. This combined sequence is then utilized as the input to the pre-trained LLM. The single concatenated embedding sequence is fed into the transformer layers of the pre-trained LLM, and all of these layers are kept frozen. At this stage, the positional encoding inherent in the LLM (e.g., RoPE in the case of Llama) is utilized directly. We have clarified the above details in Line 282-284 and Line 314-316 of the updated **PDF**.
>
> As illustrated in Line 252 and Figure 2, the task prompt is included within the text description provided to the LLM. It serves as an important guide for transforming the aligned time series embedding for the specific task. Furthermore, special tokens [start_prompt] and [end_prompt] are included at the beginning and end of the text description, respectively. This enables the LLM to effectively separate and recognize the three types of heterogeneous information. We have updated Figure 2 in the revised **PDF** to clearly illustrate the corresponding special token.
>
> **W.(b)**
>
> We have currently indicated in the paper, through both text and figures, that the prompt bank is trained during the training phase and remains frozen during the online phase. In the ‘Online Phase’ subsection of Section 4.3 of the previous PDF , we state that all parameters, except for the output projection layer, are frozen during the online phase. Additionally, Figure 2 utilizes "fire" and "ice" icons to represent whether each module is trained or frozen during the training and online phases, respectively.
>
> However, we strongly agree with the reviewer's suggestion that explicitly clarifying the status of the prompt bank (i.e., that it is frozen) during the online phase is crucial for reader comprehension, as it is a core component of our framework. We revised Line 324-326 in the updated **PDF** to address these concerns.

---

> ### Author Response · Authors · 2025-11-21
>
> **W.(c) & Q.(d)**
>
> We appreciate this practical suggestion and strongly agree that more detailed runtime statistics are required for the eventual deployment of our proposed LLM4OT. We measured specific runtime metrics for LLM4OT and existing OTSF models (i.e., FSNet, OneNet, and DSOF) across all datasets with a prediction length of 1. These metrics include the total duration of the training phase (sec), the total duration of the online phase (sec), the inference latency (sec/itr), defined as the time required per update, and the GPU memory consumption (MiB).
>
> | **FSNet** | ETTh1 | ETTh2 | ETTm1 | ETTm2 | WTH | ECL | Traffic |
> | :--- | :---: | :---: | :---: | :---: | :---: | :---: | :---: |
> | Training Phase Duration (sec) | 275 | 274 | 801 | 830 | 775 | 398 | 159 |
> | Online Phase Duration (sec) | 341 | 332 | 1,024 | 1,019 | 998 | 463 | 351 |
> | Inference Latency (sec/itr) | 0.031 | 0.030 | 0.022 | 0.022 | 0.025 | 0.023 | 0.026 |
> | GPU Memory Consumption (MiB) | 379 | 386 | 377 | 384 | 392 | 390 | 385 |
>
> | **OneNet** | ETTh1 | ETTh2 | ETTm1 | ETTm2 | WTH | ECL | Traffic |
> | :--- | :---: | :---: | :---: | :---: | :---: | :---: | :---: |
> | Training Phase Duration (sec) | 551 | 540 | 1,621 | 1,633 | 1,503 | 848 | 303 |
> | Online Phase Duration (sec) | 701 | 690 | 2,237 | 2,241 | 2,079 | 994 | 741 |
> | Inference Latency (sec/itr) | 0.063 | 0.061 | 0.050 | 0.050 | 0.051 | 0.050 | 0.053 |
> | GPU Memory Consumption (MiB) | 461 | 451 | 452 | 447 | 439 | 448 | 452 |
>
> | **DSOF** | ETTh1 | ETTh2 | ETTm1 | ETTm2 | WTH | ECL | Traffic |
> | :--- | :---: | :---: | :---: | :---: | :---: | :---: | :---: |
> | Training Phase Duration (sec) | 622 | 613 | 1,904 | 1,936 | 1,789 | 994 | 379 |
> | Online Phase Duration (sec) | 733 | 705 | 2,588 | 2,559 | 2,371 | 1,201 | 855 |
> | Inference Latency (sec/itr) | 0.064 | 0.062 | 0.058 | 0.056 | 0.057 | 0.059 | 0.062 |
> | GPU Memory Consumption (MiB) | 528 | 519 | 538 | 575 | 583 | 561 | 565 |
>
> | **LLM4OT** | ETTh1 | ETTh2 | ETTm1 | ETTm2 | WTH | ECL | Traffic |
> | :--- | :---: | :---: | :---: | :---: | :---: | :---: | :---: |
> | Training Phase Duration (sec) | 4,650 | 4,650 | 14,428 | 14,501 | 13,786 | 7,249 | 2,788 |
> | Online Phase Duration (sec) | 787 | 741 | 2,938 | 2,973 | 2,841 | 1,389 | 931 |
> | Inference Latency (sec/itr) | 0.068 | 0.065 | 0.064 | 0.066 | 0.068 | 0.065 | 0.067 |
> | GPU Memory Consumption (MiB) | 42,331 | 42,319 | 42,906 | 43,008 | 43,213 | 42,839 | 43,012 |
>
> Online Time Series Forecasting (OTSF) is a task that involves sufficiently pre-training a model on currently available data (i.e., training data) during the training phase, and subsequently adapting the model to continuously streaming data during the online phase. Therefore, it is crucial to prioritize the resources consumed during the online phase over those consumed during the training phase.
>
> The duration of the online phase and the inference latency require only slightly more time compared to existing models. This implies that despite utilizing an LLM, the model can continuously adapt to new distributions in the online phase with a significantly smaller amount of parameter learning by leveraging the LLM's inherent superior transferability along with the distribution guidance provided by the frequency basis-based pattern embedding prompting method.
>
> The utilization of the LLM results in a higher cost for both the training phase duration and GPU memory usage compared to existing models. However, the training phase is less cost-sensitive than the online phase, as it is a process of learning base knowledge through batch training on sufficient data, and thus, sufficiently acquiring knowledge is more critical. Furthermore, given the recent widespread adoption of LLM-based time series models and time series foundation models, the GPU memory utilized by LLM4OT is not considered an overly burdensome level. In conclusion, while LLM4OT requires a longer duration during the training phase and utilizes more GPU memory, we emphasize that it achieves significantly superior performance (please refer to Table 1) with only a comparable cost in the cost-sensitive online phase compared to existing models.
>
> We have revised Table 8 in Appendix F.2 of the updated **PDF** to present this experiment and provided the corresponding explanation in Line 1052-1066.

---

> ### Author Response · Authors · 2025-11-21
>
> **Q.(c)**
>
> We appreciate this meaningful question. Initially, we utilized a random initialization method for the prompt bank without employing any special initialization approach.
>
> Following the reviewer's suggestion, we have conducted an experiment on ETTh2 and Traffic datasets where the initial value of each prompt was set using the embedding from the pre-trained LLM corresponding to the words in a task-related sentence (i.e., 'Continuously adapt to shifting data distributions in online time series streams').
>
> | | H | MSE | MAE |
> | :--- | :-: | :---: | :---: |
> | **ETTh2** | 1 | 0.416 | 0.435 |
> | | 24 | 0.941 | 0.612 |
> | | 48 | 1.247 | 0.694 |
> | **Traffic** | 1 | 0.231 | 0.306 |
> | | 24 | 0.333 | 0.327 |
> | | 48 | 0.359 | 0.419 |
>
> Utilizing the pre-trained LLM's token embedding for task-related words for prompt initialization yields performance that is either comparable to or even worse than the conventional random initialization shown in Table 1. However, regardless of the initialization method employed, our approach demonstrates significantly superior performance compared to existing baselines. This provides strong evidence for the effectiveness of a prompting method-based distribution guide within the OTSF scenario.
>
> Furthermore, the learned prompts are trained to share the embedding space with the tokens of the pre-trained LLM. In this training process, where only the prompt vectors are updated while the LLM remains frozen, the vector values are optimized so that the fixed LLM produces the desired output upon receiving them. Consequently, these prompts become functionally aligned in a manner that is comprehensible to the LLM's attention mechanism.
>
> We emphasize again that the main focus of this study is not on factors such as the initialization method of the prompting technique, but rather on demonstrating an improvement in both effectiveness and efficiency by applying a frequency basis-based prompting method to leverage the knowledge of a pre-trained LLM without further training to continuously counter distribution shifts. While this research proves the effectiveness of the prompting method, at least in a dynamic (i.e., online) scenario, the prompting method remains underexplored for time series forecasting tasks in both static and dynamic settings. Therefore, as suggested by the reviewer, we consider future work to include further exploration into initialization techniques or more sophisticated prompt and LLM space alignment methods to develop a framework that is even more robust to various distribution shifts prevalent in time series tasks.

---

### Official Review · Reviewer_9hb9 · 2025-10-29

**Soundness:** 2
**Presentation:** 2
**Contribution:** 2
**Rating:** 4
**Confidence:** 5

**Summary:**

This paper proposes an online learning method based on LLM and frequency component activation.

**Strengths:**

1、Introducing LLM into online learning is a rather interesting approach.

**Weaknesses:**

1、Why are the results corresponding to the blue modules in Figures 1a and 1b the same? Is the data segmentation ratio in 1a consistent with that in 1b?

2、The definition of cross-domain prediction is not clearly defined. If ETTh1 is used as the training set and ETTh2 is used as the online set, is it still necessary to perform online learning based on the data of ETTh2? Or should we only train on ETTh1 and perform inference directly on ETTh2?

3、Frequency domain decomposition operations and operations based on frequency components as basis vectors are quite common in the time series domain.

4、A prediction length of 1 is rare in real-world scenarios, and is updating part of the model's parameters based on each sample too frequent? How can the time interval between model inference calls be balanced with the frequency of model updates?

5、Cross-domain forecasting recommendations are compared with a range of current time series baseline models, including zero-shot and few-shot operations.

6、The definitions of online learning of time series and time series forecasting are not very clear to me, and the description in this article makes it even more difficult for me to understand.

**Questions:**

See Weaknesses.

---

> ### Author Response · Authors · 2025-11-21
>
> **W1.**
>
> The results of the blue modules in Figures 1a and 1b are identical. Both blue modules represent the performance of each model under the default setting (i.e., a 20%/5%/75% split ratio and an ETTh2→ETTh2 scenario).
>
> Figure 1a presents an experiment to observe robustness against pattern shift. It compares the standard ETTh2→ETTh2 experiment (i.e., blue) with the ETTh1→ETTh2 cross-domain experiment (i.e., red), while maintaining the default 20%/5%/75% split ratio.
>
> Figure 1b presents an experiment to observe robustness in an extended online phase. It compares the default 20%/5%/75% split ratio case (i.e., blue) with a 10%/5%/85% split ratio case (i.e., red), while maintaining the ETTh2→ETTh2 dataset setting.
>
> Therefore, the blue modules in Figures 1a and 1b correctly represent the same experimental results.
>
> **W2.**
>
> In the OTSF task, the cross-domain scenario does not involve performing zero-shot inference on the ETTh2 dataset after training on the ETTh1 dataset. Instead, as specified in Line 64-65, the scenario involves training the model on the ETTh1 dataset and subsequently performing online updates on the ETTh2 dataset as it arrives in a stream. Therefore, the former of the two cases mentioned by the reviewer is correct.
>
> Consistent with prior studies [1,2,3] and our own research, the scenario for the OTSF task involves sufficiently pre-training a model on the currently available dataset. Then, as additional data continuously streams in, the pre-trained model is adapted to the new data using only fast and lightweight updates. Consequently, the objective of our task differs from one that solely performs zero-shot inference on an online set after the initial training phase.
>
> [1] Pham, Quang, et al. "Learning fast and slow for online time series forecasting." arXiv preprint arXiv:2202.11672 (2022).
>
> [2] Wen, Qingsong, et al. "Onenet: Enhancing time series forecasting models under concept drift by online ensembling." Advances in Neural Information Processing Systems 36 (2023): 69949-69980.
>
> [3] Lau, Ying-yee Ava, Zhiwen Shao, and Dit-Yan Yeung. "Fast and Slow Streams for Online Time Series Forecasting Without Information Leakage." The Thirteenth International Conference on Learning Representations. 2025.
>
> **W3.**
>
> While numerous studies have leveraged frequency-domain representations, they have all been conducted in static scenarios. We emphasize that this is the first attempt to enable the LLM to adapt to distribution shifts, leveraging time series patterns analyzed via fourier domain modeling in a dynamic setting. The main challenge in the Online Time Series Forecasting (OTSF) scenario is to continually adapt to evolving distributions using only limited data.
>
> Directly fine-tuning an LLM for robustness against distribution shifts incurs significant computational cost, making it unsuitable for the OTSF scenario. Therefore, rather than tuning the large-scale model directly, we sought to guide the LLM by learning small, distinct soft prompts for each distribution via a prompting method. However, the strategy of incrementally tuning prompts for every new, unseen distribution is ineffective. To address this, we explored the underlying basis of time series patterns, leading us to propose a novel approach grounded in a frequency basis. By utilizing Fourier decomposition, the time series is decomposed into its underlying frequency bases. We then learn the knowledge of each basis as a prompt. Consequently, during the online phase, even upon the emergence of unseen patterns, the LLM can be guided by leveraging combinations of this pre-learned, frozen basis distribution knowledge (i.e., the prompts for each basis) without requiring any additional training.
>
> In summary, while our method originates from existing techniques, we wish to emphasize that we did not merely adopt them as-is. Instead, we devised a novel and efficient prompting method specifically tailored for the OTSF scenario, which is characterized by data scarcity and demands high efficiency.

---

> ### Author Response · Authors · 2025-11-21
>
> **W4.**
>
> We thank the reviewer for the insightful discussion. As the reviewer noted, the task of predicting only the immediate next time step (prediction length = 1) may not be perceived as practical, and updating the model at every time point can be inefficient. However, we would like to clarify that the aforementioned experimental setting was utilized to maintain consistency, as it is widely accepted within the online time series forecasting community.
>
> However, as we are open to suggestions, we investigated the trade-off between performance and efficiency (i.e., Online Update Time) that arises when adjusting the model update frequency during the online phase. Using the ETTh2 dataset with a prediction length of 24 and 48, we set the model update frequencies to {1, 2, 4, 6, 12, 24}.
>
> | H=24 | Update Frequency | MSE | MAE | Online Update Time (sec) |
> | :--- | :---: | :---: | :---: | :---: |
> | | 1 | 0.932 | 0.607 | 811 |
> | | 2 | 0.949 | 0.614 | 387 |
> | | 4 | 1.005 | 0.640 | 192 |
> | | 6 | 1.096 | 0.677 | 111 |
> | | 12 | 1.283 | 0.762 | 58 |
> | | 24 | 1.353 | 0.789 | 30 |
>
> | H=48 | Update Frequency | MSE | MAE | Online Update Time (sec) |
> | :--- | :---: | :---: | :---: | :---: |
> | | 1 | 1.245 | 0.699 | 837 |
> | | 2 | 1.255 | 0.705 | 395 |
> | | 4 | 1.309 | 0.726 | 202 |
> | | 6 | 1.368 | 0.754 | 111 |
> | | 12 | 1.544 | 0.832 | 57 |
> | | 24 | 1.586 | 0.848 | 29 |
>
> By investigating the relationship between model update frequency and online performance, we confirmed a clear trade-off: decreasing the update frequency during the online phase led to a noticeable reduction in online update time alongside a slight degradation in predictive performance. Although this minor performance drop is observed compared to updating every step (i.e., Update Frequency = 1), our model maintains performance that is two times or more superior to the baselines listed in Table 1, even when the update frequency is extended up to 24. Furthermore, by setting the update frequency to be just 2x rarer, the model records an overall online phase duration that is either faster than or comparable to all existing OTSF baselines (please refer to Appendix F.2).
>
> In conclusion, we maintained the conventional experimental setting utilized in prior OTSF studies. However, if we place a greater focus on model efficiency, we discovered that slightly reducing the update frequency during the online phase is feasible. This approach ensures excellent efficiency while maintaining significantly superior performance compared to other baselines, demonstrating that LLM4OT is more suitable for practical scenarios.

---

> ### Author Response · Authors · 2025-11-21
>
> **W5.**
>
> We appreciate the insightful suggestion. Although Table 3 currently compares LLM4OT only against existing OTSF baselines for the cross-domain scenario, we have extended the baselines and performed a comparison with the time series baselines (i.e., DLinear, PatchTST, and iTransformer) that were utilized in Table 1. The experimental setting is the same as in Table 3.
>
> | Training | Online | DLinear (MSE) | DLinear (MAE) | PatchTST (MSE) | PatchTST (MAE) | iTransformer (MSE) | iTransformer (MAE) | Online |  DLinear (MSE) | DLinear (MAE) | PatchTST (MSE) | PatchTST (MAE) | iTransformer (MSE) | iTransformer (MAE)
> | :--- | :---: | :---: | :---: | :---: | :---: | :---: | :---: | :---: | :---: | :---: | :---: | :---: | :---: | :---: |
> | ETTh1 | ETTh2 | 0.921 | 0.940 | 1.773 | 1.254 | 1.798 | 1.241 | ETTm2 | 0.811 | 0.801 | 1.621 | 1.189 | 1.917 | 1.326 |
> | ETTh2 | ETTh1 | 1.042 | 0.920 | 1.838 | 1.277 | 1.047 | 0.938 | ETTm2 | 1.034 | 0.917 | 1.681 | 1.215 | 1.795 | 1.307 |
> | ETTm1 | ETTh2 | 1.136 | 0.969 | 1.886 | 1.273 | 1.931 | 1.298 | ETTm2 | 0.842 | 0.878 | 1.372 | 1.134 | 1.484 | 1.181 |
> | ETTm2 | ETTh2 | 1.204 | 1.028 | 1.839 | 1.284 | 1.897 | 1.297 | ETTm1 | 1.211 | 1.004 | 1.695 | 1.249 | 1.879 | 1.326 |
>
>
> We made the following observations:
>
> - The time series baseline models exhibit superior performance in the cross-domain scenario, with the exception of the existing state-of-the-art DSOF and our proposed LLM4OT. This result suggests that models like FSNet and OneNet did not truly acquire the underlying structure of the newly arriving patterns, but rather exploited leaked data during their online phase.
>
> - However, we confirm a significant performance drop for all models compared to the single-domain experiments presented in Table 1. This indicates that simple tuning alone is insufficient for models to quickly adapt to the distribution shift caused by the limited amount of streaming data. Nevertheless, DLinear exhibits superior performance compared to PatchTST and iTransformer, suggesting that simpler linear transformation-based methodologies adapt better to new distributions with minimal tuning than complex transformer-based ones.
>
> - LLM4OT exhibits superior adapting performance compared to time series baselines, not only in Table 1 but also in the cross-domain scenario. This indicates that the combination of the LLM’s excellent zero/few-shot ability, the data augmentation provided by the text descriptions, and the frequency basis prompting methodology aids in rapid adaptation within data-scarce online learning scenarios.
>
> **W6.**
>
> This task is analogous to Continual Learning [1,2,3], which utilizes a single model to learn sequential tasks, prevent catastrophic forgetting, and maximize adaptation quality to new tasks. However, due to the inherent characteristics of time series data, discretely dividing it into tasks as is done in continual learning is ambiguous. Consequently, in the time series domain, the online learning task is widely employed, which involves continuously and frequently updating the model as new data instances arrive at each time step.
>
> Online Time Series Forecasting (OTSF) is a scenario that involves two phases: an initial training phase, where a model is sufficiently pre-trained on a currently available dataset, and an online phase, where the pre-trained model is adapted to new data at every iteration as additional data continuously streams in.
>
> Accordingly, the main challenge in the OTSF scenario is to quickly adapt the model to new distributions using only lightweight updates, especially when the distribution of the dataset streaming in during the online phase continuously shifts relative to the training phase.
>
> To enhance the understanding of the task addressed in this study, we will provide a more precise and elaborate explanation of its meaning throughout the paper.
>
> [1] Rebuffi, Sylvestre-Alvise, et al. "icarl: Incremental classifier and representation learning." Proceedings of the IEEE conference on Computer Vision and Pattern Recognition. 2017.
>
> [2] Tiwari, Rishabh, et al. "Gcr: Gradient coreset based replay buffer selection for continual learning." Proceedings of the IEEE/CVF Conference on Computer Vision and Pattern Recognition. 2022.
>
> [3] Yoon, Jaehong, et al. "Online coreset selection for rehearsal-based continual learning." arXiv preprint arXiv:2106.01085 (2021).

---

### Official Review · Reviewer_vTbY · 2025-11-01

**Soundness:** 2
**Presentation:** 3
**Contribution:** 3
**Rating:** 4
**Confidence:** 3

**Summary:**

This paper proposes LLM4OT, a framework for online time series forecasting that integrates large language models with frequency-domain pattern recognition. It introduces frequency-driven prompts to encode transferable temporal patterns and an LLM-enhanced adaptation module that converts recent time-series trends into textual descriptions for semantic reasoning. By combining spectral decomposition with LLM priors, LLM4OT achieves robust forecasting under distribution shifts and long-term online scenarios, outperforming existing state-of-the-art methods on multiple real-world benchmarks.

**Strengths:**

The paper is well-written and clearly organized, with thorough explanations of both the model design and the motivation behind each component. The methodology is presented in a way that is easy to follow, with clear distinctions between the training and online adaptation phases.

The experimental evaluation is extensive and convincing, covering multiple real-world datasets, diverse baselines (including traditional, online, and LLM-based models), and comprehensive ablation studies. These results strongly support the effectiveness and robustness of the proposed LLM4OT framework.

Overall, while the conceptual novelty lies mainly in combining frequency-domain prompting with LLM-based reasoning, the work distinguishes itself through its systematic empirical validation, detailed analysis, and clarity of presentation, making it a solid and well-executed contribution to online time series forecasting.

**Weaknesses:**

The primary weakness of this paper lies in its limited conceptual novelty. Both of the core ideas—leveraging frequency-domain representations and incorporating LLMs for sequence modeling—have been explored in prior works. While the proposed combination of frequency-driven prompting and LLM-based adaptation is practically useful, the paper does not sufficiently justify why these two components need to be integrated or what complementary benefits the frequency domain provides to LLM reasoning. The argument that frequency prompts help encode transferable temporal patterns remains mostly empirical, without deeper theoretical or analytical discussion. A stronger motivation or ablation directly comparing LLM4OT to time-domain prompting or alternative representation schemes (e.g., wavelet or trend-seasonality decomposition) would help clarify this design choice.

Moreover, the paper lacks a direct comparison with strong recent baselines, particularly the TimeMixer series, which are currently leading architectures for both static and online forecasting. Since these models also capture multi-scale temporal dependencies and exhibit strong generalization under distribution shifts, omitting them makes it difficult to assess the relative advancement of LLM4OT. Including such comparisons, or at least providing analytical discussion of the differences in design and performance trade-offs, would make the empirical evaluation more convincing.

Overall, while the work is well-executed and experimentally thorough, it would benefit from a clearer theoretical justification for the use of frequency-domain prompting with LLMs and a more comprehensive comparison against state-of-the-art forecasting frameworks to better position its originality and impact.

**Questions:**

The paper proposes combining (a) frequency-based pattern embeddings derived from DFT/DWT and (b) an LLM with text descriptions of recent behavior. Conceptually, both ingredients individually have precedents in the literature (frequency-domain decomposition for forecasting, and LLM-augmented forecasters). Could you elaborate on why these two components must be used together and why this specific pairing is essential for online adaptation? In other words: is there a principled reason that an LLM cannot adapt equally well using only time-domain representations and recent numeric context, without frequency prompts?

---

> ### Author Response · Authors · 2025-11-21
>
> **W1. & Q1. (1)**
>
> While numerous studies have incorporated LLMs for sequence modeling or leveraged frequency-domain representations, they have all been conducted in static scenarios. First of all, we emphasize that this is the first attempt to enable the LLM to adapt to distribution shifts, leveraging time series patterns analyzed via fourier domain modeling in a dynamic setting.
>
> The main challenge in the Online Time Series Forecasting (OTSF) scenario is to continually adapt to evolving distributions using only limited data. To align the embedding generated by the time series backbone network continuously with the shifted distribution in the constrained scenario described above, we utilize a pre-trained LLM. The superior generalizability and transferability of the LLM enable the model to adapt quickly and consistently in practical scenarios where the online phase is prolonged (i.e., Figure 1(c)), even with limited data and updates. To further enhance the LLM's adapting capability, we utilize the following two components: (a) text descriptions for data augmentation, and (b) frequency-based pattern embeddings for robustness against distribution shift.
>
> To amplify the limited data, we leverage text descriptions. The effectiveness of this approach, injecting recent pattern information by additionally utilizing the text modality (i.e., text descriptions) in addition to time series data, is validated in the ablation study in Table 2. Notably, this result is achieved without any training process.
>
> Directly fine-tuning an LLM for robustness against distribution shifts incurs significant computational cost, making it unsuitable for the OTSF scenario. Therefore, rather than tuning the large-scale model directly, we sought to guide the LLM by learning small, distinct soft prompts for each distribution via a prompting method. However, a criterion for partitioning time series distributions is necessary, and the strategy of incrementally tuning prompts for every new, unseen distribution is ineffective. To achieve this, we explored a frequency domain-grounded approach instead of the time domain. While the time domain allows for distribution discrimination via techniques like clustering based on time series representations, this process is vulnerable to distribution shift and requires a dedicatedly trained time series encoder. Conversely, the frequency domain enables distribution discrimination through Fourier transformation without any training and inherently provides robustness to distribution shift, which allows for effective handling of unseen distribution. Specifically, by utilizing Fourier decomposition, the time series is decomposed into its underlying frequency bases. We then learn the knowledge of each basis as a prompt. Consequently, during the online phase, even upon the emergence of unseen patterns, the LLM can be guided by leveraging combinations of this pre-learned, frozen basis distribution knowledge (i.e., the prompts for each basis) without requiring any additional training.
>
> We wish to emphasize that while these are individually existing approaches, we did not merely adopt them as-is. Rather, we devised a novel and efficient prompting method and a strategy for providing time-related recent pattern information via text description. Both approaches are specifically tailored for the OTSF scenario, which is characterized by data scarcity and demands high efficiency.

---

> ### Author Response · Authors · 2025-11-21
>
> **W1. & Q1. (2)**
>
> Additionally, we conducted a comparison with a time-domain prompting method using ETTh2 dataset to demonstrate that our frequency basis-based approach captures underlying time series patterns. Our objective in applying the prompting method to the OTSF scenario is to assign distinct prompts to time series data originating from different distributions, thereby enabling the large model to adapt to diverse distributions without tuning. In the time domain, we utilize a method that classifies time series distributions by performing clustering based on the representations of the training data, and subsequently learning a distinct prompt for each cluster. For this, we employ a time series backbone network pre-trained on the training data to generate reliable representations for clustering, and the mean representation of each cluster is set as its prototype. For time series emerging in the online phase, the prompt corresponding to the closest prototype's cluster is then utilized.
>
> We conducted experiments comparing the frequency domain-based prompting method and the time domain-based prompting method using the following two settings.
>
> - Case 1:Learning prompts only during the training phase and then utilizing the prompts while they are frozen during the online phase (i.e., LLM4OT version).
>
> - Case 2: Continuously learning the prompts in both the training and online phases (i.e., Optimal version).
>
> | H \ Frequency (Case 1) | MSE | MAE | Online Update Time (sec) | H \ Time (Case 1) | MSE | MAE | Online Update Time |
> | :---: | :---: | :---: | :---: | :---: | :---: | :---: | :---: |
> | 1 | 0.425 | 0.601 | 741 | 1 | 0.635 | 0.756 | 843 |
> | 24 | 1.217 | 0.993 | 811 | 24 | 1.597 | 1.076 | 931 |
> | 48 | 1.482 | 1.073 | 837 | 48 | 1.924 | 1.273 | 958 |
>
> | H \ Frequency (Case 2) | MSE | MAE | Online Update Time (sec) | H \ Time (Case 2) | MSE | MAE | Online Update Time |
> | :---: | :---: | :---: | :---: | :---: | :---: | :---: | :---: |
> | 1 | 0.392 | 0.586 | 936 | 1 | 0.395 | 0.598 | 1,031 |
> | 24 | 1.017 | 0.908 | 1,028 | 24 | 1.101 | 1.001 | 1,141 |
> | 48 | 1.277 | 0.944 | 1,330 | 48 | 1.263 | 0.923 | 1,438 |
>
> In both domains, utilizing the approach of continuously learning the prompts in the online phase (i.e., Case 2) records superior performance, but the online update process is not efficient. When utilizing the prompt based on the time domain, we observe a significant performance drop when the prompt is learned only during the training phase and then utilized while frozen in the online phase (i.e., Case 1). This indicates that the representation clustering-based distribution classification method used in the time domain fails to perform effective classification when a shift occurs. Conversely, the prompting method utilizing the frequency domain effectively represents the new distribution as a combination of underlying frequency bases, allowing it to maintain performance even when a shift occurs.
>
> Additionally, we confirmed this difference from the perspective of the learned prompt itself. We compared the similarity between the prompt finalized after the end of the training phase in Case 1 and the prompt finalized after the online phase was completed in Case 2, across both domains.
>
> | | Frequency Domain | Time Domain |
> | :---: | :---: | :---: |
> |Similarity | 0.7375 | 0.2481 |
>
> Compared to the time domain, the frequency domain prompt exhibits a significantly higher similarity between the learned prompts in both cases. This demonstrates that the effective learning of the underlying basis during the training phase alone enables the model to adequately handle the shift without additional learning in the online phase.

---

> ### Author Response · Authors · 2025-11-21
>
> **W2.**
>
> As suggested by the reviewer, we compared the performance of the TimeMixer[1] against LLM4OT under the experimental settings of Table 1.
>
> |  | H | MSE | MAE |
> | :--- | :-: | :---: | :---: |
> | **ETTh1** | 1 | 0.557 | 0.706 |
> | | 24 | 2.719 | 1.528 |
> | | 48 | 3.274 | 1.699 |
> | **ETTh2** | 1 | 0.439 | 0.563 |
> | | 24 | 1.938 | 1.296 |
> | | 48 | 2.216 | 1.398 |
> | **ETTm1** | 1 | 0.137 | 0.341 |
> | | 24 | 0.602 | 0.726 |
> | | 48 | 0.811 | 0.882 |
> | **ETTm2** | 1 | 0.114 | 0.326 |
> | | 24 | 0.525 | 0.688 |
> | | 48 | 0.784 | 0.854 |
> | **WTH** | 1 | 0.338 | 0.543 |
> | | 24 | 1.208 | 1.002 |
> | | 48 | 1.733 | 1.216 |
> | **ECL** | 1 | 2.708 | 1.549 |
> | | 24 | 7.209 | 2.583 |
> | | 48 | 9.243 | 2.840 |
> | **Traffic** | 1 | 0.280 | 0.509 |
> | | 24 | 0.661 | 0.796 |
> | | 48 | 0.759 | 0.856 |
>
> We made the following observations.
>
> - The TimeMixer methodology outperforms other models, excluding the existing state-of-the-art DSOF and our proposed LLM4OT. This effectiveness stems from TimeMixer's use of multiscale information via simple linear transformation, which prevents the model from overfitting to noise or short-term variations and enables it to focus on learning the fundamental temporal patterns common across time series data.
>
> - LLM4OT outperforms the TimeMixer baseline. While TimeMixer excels at learning general time series patterns, it is not designed to adapt to a new distribution in only a few steps using limited data, which is characteristic of an online scenario. In contrast, LLM4OT distinguishes itself in the online scenario by leveraging the LLM’s inherent generalizability and excellent zero/few-shot ability. Furthermore, it utilizes text descriptions to augment limited data and employs frequency basis pattern embeddings as prompts to enable adaptation to new distributions with minimal tuning.
>
> We have revised Table 1 of the updated **PDF** to reflect the above results.
>
> [1] Wang, Shiyu, et al. "Timemixer: Decomposable multiscale mixing for time series forecasting." arXiv preprint arXiv:2405.14616 (2024).

---

### Official Review · Reviewer_JaXs · 2025-11-01

**Soundness:** 2
**Presentation:** 2
**Contribution:** 2
**Rating:** 2
**Confidence:** 3

**Summary:**

The authors improve upon current online time series forecasting methods by representing time series as combination of frequency bases and connecting a pretrained LLM with a time series backbone.

**Strengths:**

- original idea
- reasonably well written
-

**Weaknesses:**

- unclear integration and roles of LLM and time series backbone
- unclear handling of distribution shifts occuring over time in the online period
- very long supplementary material, making the paper itself not self-contained

**Questions:**

Which roles are played by the LLM and by the time series backbone?
Do you handle distribution shifts in the online period and how?

---

> ### Author Response · Authors · 2025-11-21
>
> **W1. & Q1.**
>
> The primary challenge of the Online Time Series Forecasting (OTSF) task is to enable a model to efficiently and effectively adapt to a data distribution that changes at every time step, while simultaneously retaining the knowledge acquired during the training phase. To address each of these challenges, we utilize a time series backbone and a large language model (LLM) in conjunction.
> The time series backbone serves as the most fundamental encoder for understanding time series data. It is trained during the training phase to enable the entire framework to interpret time series data. However, the time series backbone alone cannot adequately respond to distribution shifts.
>
> Therefore, we utilize a pre-trained LLM in conjunction. The primary role of the LLM in LLM4OT is to calibrate the time series embeddings generated by the time series backbone to align with the current distribution. The superior generalizability and transferability of the LLM enable the model to adapt quickly and consistently in practical scenarios where the online phase is prolonged (i.e., Figure 1(c)), even with limited data and updates.
>
> In conclusion, the time series backbone enables the entire framework to interpret time series data and is slowly learned during the training phase. In contrast, to perform the quick adaptation essential for online learning, the pre-trained LLM utilizes the pattern embedding (which acts as a prompt) and the text description embedding (which contains supplementary information in data-limited scenarios) in conjunction with the time series embedding generated by the time series backbone. This allows the model to respond to continuous distribution shifts.
>
>
> **W2.**
>
> To maximize the LLM's capability to counter distribution shifts during the online phase with minimal training, we propose a novel prompting method based on a frequency basis. Existing specific prompting methods [1,2] have the advantage of enabling a large backbone model to adapt to diverse distributions without tuning by assigning distinct prompts. However, they possess a significant drawback: new prompts must be continually learned when there are continuous distribution shifts. As even prompt tuning can be computationally burdensome in the online phase, we trained prompts to represent each basis of the time series patterns, derived through frequency basis decomposition. This new approach enables continuous adaptation to distribution shifts. Even if an unseen distribution is encountered during the online phase, it can be represented by a combination of these basis prompts without requiring any additional training.
>
> Although frequency analysis has been widely utilized in the time series domain, its application has been confined to static scenarios. Our work marks its first application in a dynamic online learning context. Furthermore, this is the first attempt to utilize frequency basis prompting to enable the model to discern data distributions during the online phase without any online training, relying solely on knowledge from the training phase. This approach is highly suitable for online learning, which must be cost-effective (please refer to Appendix F.2).
>
> In summary, LLM4OT is the first to discover that the superior generalizability and transferability of LLMs are effective for adapting to new distributions quickly in an online scenario, even with limited data and updates. Furthermore, to adapt the model to distribution shifts with minimal training during the online phase, we developed and applied a frequency basis prompting method to resolve this challenge.
>
> [1] Wang, Yabin, Zhiwu Huang, and Xiaopeng Hong. "S-prompts learning with pre-trained transformers: An occam’s razor for domain incremental learning." Advances in Neural Information Processing Systems 35 (2022): 5682-5695.
>
> [2] Wang, Zifeng, et al. "Dualprompt: Complementary prompting for rehearsal-free continual learning." European conference on computer vision. Cham: Springer Nature Switzerland, 2022.

---

> ### Author Response · Authors · 2025-11-21
>
> **W3.**
>
> We aimed to enhance reader comprehension by providing extensive supplementary material. This material includes more detailed explanations of each module within LLM4OT, as well as various experiments conducted to demonstrate its effectiveness and efficiency.
>
> As the supplementary material is, by definition, auxiliary, we have structured the main body of the paper to be self-contained. It provides a sufficient understanding of the core concepts of LLM4OT even without reference to the supplementary content.
> Therefore, we respectfully request that the paper’s presentation quality not be judged negatively solely based on the length of the supplementary material. We are, however, prepared to provide clear explanations for any specific points of concern.

---

### Author Response · Authors · 2025-12-02
**Summary of Rebuttal and Key Experimental Results**

We are grateful for all the constructive feedback from the reviewers and look forward to further discussion to address any remaining questions and clarify potential misunderstandings regarding our contribution.

The primary concerns raised by the reviewers are summarized as follows.

**(1)** The utilization of LLMs and frequency components in time series forecasting are pre-existing techniques.

**(2)** Sufficient justification for why pattern embedding should be performed in the frequency domain rather than the time domain

**(3)** Inference Cost

In light of this feedback, we carried out comprehensive experiments that directly resolve the raised concerns.

**- Differentiated Strategy for Utilizing LLM and Frequency Components** (addressed in **[Reviewer vTbY] W1, [Reviewer 9hb9] W3**)

We have clearly explained that our approach, utilizing both the LLM and frequency component in a dynamic (i.e., online) scenario, is the first of its kind. We observed that the LLM's transferability is beneficial for the long-persisting online phase, and explained that our frequency basis-based prompting method, designed to minimize the online update cost, is both technically and conceptually novel.

**- Rationale for Generating Frequency-based Pattern Embeddings** (addressed in **[Reviewer vTbY] Q1**)

We compared the time domain-based prompting method with our proposed frequency domain-based prompting method. The results demonstrate that the pattern embeddings generated based on the frequency domain effectively learn the basis of time series patterns, allowing them to continuously guide the LLM's distribution even when frozen during the online phase, using only the knowledge acquired during the training phase.

**- Inference Cost** (addressed in **[Reviewer Sjmn] W(c) & Q(d)**)

We provided runtime statistics for all datasets utilized in our experiments, considering the total duration of the training phase (sec), the total duration of the online phase (sec), the inference latency (sec/itr), and GPU memory consumption (MiB). While the utilization of the LLM necessitates higher memory consumption and a longer duration in the training phase, we confirmed that the online phase duration and inference latency remain comparable to existing OTSF models. This demonstrates that if LLM4OT undergoes sufficient training in the initial phase, it can adapt the model quickly and without delay when data continuously streams in during the online phase, thereby achieving substantially higher performance compared to existing models.

Additionally, we have fully addressed all concerns and questions raised by the reviewers and incorporated their constructive suggestions into the updated PDF. These results show that LLM4OT effectively leverages the LLM’s transferability in the online scenario and achieves significantly superior performance through efficient adaptation via minimal parameter updates in the online phase, utilizing a novel frequency-based prompting method. We are confident that these comprehensive experiments and explanations have resolved the reviewers’ concerns.

---

Unfortunately, due to the incident occurring during the rebuttal period, our reviewers had insufficient time to fully evaluate our responses before the discussion period concluded. Under normal circumstances, we believe that substantive discussion would have directly addressed all reviewer concerns.

We therefore urge you to confirm the thoroughness of our responses when formulating your recommendation. We thank you for your careful consideration.

---

### Meta-Review · Area_Chair_tES3 · 2025-12-22

**Summary:**

This paper proposes **LLM4OT**, a framework for online time series forecasting that leverages a frozen large language model together with frequency-domain pattern prompting to handle distribution shift. Reviewers raised concerns regarding the **true methodological novelty**, the **necessity of using frequency-domain prompting and LLMs**, and the **practicality and scalability** of the approach in realistic online settings. While the rebuttal provides additional explanations, ablation studies, and efficiency analyses, these responses primarily **clarify the design choices rather than fundamentally strengthening the core contribution**. After considering the discussion, I believe that the paper still falls short in terms of novelty and conceptual depth relative to the acceptance bar.

**Reviewer Concerns:**

- **Partially addressed concerns:**
  - The rebuttal improves clarity regarding the division of labor between the time-series backbone and the frozen LLM, and provides additional experimental comparisons (e.g., with TimeMixer).
  - Runtime and memory statistics are added, which help assess feasibility in online settings.
- **Remained concerns:**
  - **Limited novelty (multiple reviewers):**
    The framework largely combines existing components, frequency-domain analysis, prompt-based LLM adaptation, and standard OTSF backbones, without introducing fundamentally new modeling principles or theoretical insights. The rebuttal does not convincingly demonstrate that the proposed integration constitutes a sufficiently novel methodological advance.
  - **Justification of frequency-domain prompting:**
    Although additional experiments are provided, the advantage of frequency-domain prompts over carefully designed time-domain or lightweight adaptive modules remains incremental. The gains do not clearly justify the added complexity.
  - **Use of large LLMs in OTSF:**
    Despite being frozen, reliance on large LLMs significantly increases system complexity. The rebuttal acknowledges this limitation, but does not fully alleviate concerns regarding deployability and general applicability.
  - **Empirical improvements are moderate:**
    Performance gains over strong baselines are relatively small and sometimes inconsistent across datasets, making it difficult to conclude that the method provides a robust improvement rather than a scenario-specific enhancement.

Overall, the rebuttal improves presentation as well as clarifying some technical details for helping understanding the solutions, but does not sufficiently resolve the main conceptual concerns raised by the reviewers.

**Reviewer Scores:**

This manuscript receives four comments with consistent negative evaluations. The discussion phase did not involve substantial back-and-forth between the reviewers and the authors. With the initial consistent negative scores, I have to make this decision of rejection on this current version.

---

### Decision · Program_Chairs · 2026-01-26

Reject